# LET-381/FoxF and its target UNC-30/Pitx2 specify and maintain the molecular identity of *C. elegans* mesodermal glia that regulate motor behavior

Nikolaos Stefanakis (ID)[1], Jessica Jiang (ID)[1], Yupu Liang[2,3] & Shai Shaham (ID)[1✉]

## Abstract

While most glial cell types in the central nervous system (CNS) arise from neuroectodermal progenitors, some, like microglia, are mesodermally derived. To understand mesodermal glia development and function, we investigated *C. elegans* GLR glia, which envelop the brain neuropil and separate it from the circulatory system cavity. Transcriptome analysis shows that GLR glia combine astrocytic and endothelial characteristics, which are relegated to separate cell types in vertebrates. Combined fate acquisition is orchestrated by LET-381/FoxF, a fate-specification/maintenance transcription factor also expressed in glia and endothelia of other animals. Among LET-381/FoxF targets, the UNC-30/Pitx2 transcription factor controls GLR glia morphology and represses alternative mesodermal fates. LET-381 and UNC-30 co-expression in naive cells is sufficient for GLR glia gene expression. GLR glia inactivation by ablation or *let-381* mutation disrupts locomotory behavior and promotes salt-induced paralysis, suggesting brain-neuropil activity dysregulation. Our studies uncover mechanisms of mesodermal glia development and show that like neuronal differentiation, glia differentiation requires autoregulatory terminal selector genes that define and maintain the glial fate.

**Keywords** Glia Development; *let-381*; Locomotory Behavior; Terminal Selector; *unc-30*
**Subject Categories** Chromatin, Transcription & Genomics; Development; Neuroscience

## Introduction

Glia are abundant and diverse cellular components of most, if not all, nervous systems, and are anatomically positioned to affect every aspect of signal transduction and processing in the brain. Glia dynamically regulate neuronal activity in response to presynaptic cues, provide insulation around axons and at synapses, and supply trophic support for neuron survival (Allen and Lyons, 2018). Most glia, including astrocytes and myelinating glia, are derived from neuroectodermal precursors (Kastriti and Adameyko, 2017; Rowitch and Kriegstein, 2010). By contrast, other glia, such as microglia, which are born in the yolk sac and migrate into the CNS, arise from mesodermal progenitors (Ginhoux et al, 2010; Ginhoux et al, 2013; Ginhoux and Prinz, 2015). Transcription factors regulating the development of some neuroectodermal glia are known (Hochstim et al, 2008; Rowitch and Kriegstein, 2010; Wegner, 2020); however, less is understood about the control of mesodermal glia differentiation. Furthermore, only a few factors sufficient to confer specification of certain glia subtypes in naive cellular settings are known (Canals et al, 2018).

The nematode *C. elegans* has been instrumental in uncovering basic principles of glia development and function (Shaham, 2015a). The nervous system of the adult *C. elegans* hermaphrodite contains 56 glial cells. 50 of these derive from the AB blast cell lineage, which also produces neurons and epithelial cells. Six GLR glia derive from the MS blastomere, which primarily generates body wall and pharyngeal muscle (Fig. 1A) (Sulston et al, 1983). Thus, as in vertebrates, *C. elegans* possesses glia of both neuroectodermal and mesodermal origin. Some genes affecting *C. elegans* neuroepithelial glia development have been characterized (Mizeracka et al, 2021; Shaham, 2015b; Wallace Sean et al, 2016; Zhang et al, 2020); however, virtually nothing is known about GLR glia development and functions. GLR glia extend intricate, non-overlapping sheet-like processes that ensheath the inner aspect of the *C. elegans* brain neuropil (the nerve ring; Fig. 1B) and that are adjacent to neuromuscular synapses (White et al, 1986). At the nerve ring, GLR glia are electrically coupled to the RME motoneurons through gap junctions (White et al, 1986), and uptake extracellular GABA (Gendrel et al, 2016). More anteriorly, GLR glia extend thin processes that fasciculate with sensory neuron dendrites. GLR glia also physically separate the nerve ring from the pseudocoelomic body cavity which surrounds the pharynx and acts as a rudimentary circulatory system. The proximity to synapses and to the circulatory system, the association with GABA, and the ability to phagocytose injured neurons (Altun and Hall, 2016; Nass et al, 2002; White et al, 1986) make comparisons between GLR glia and astrocytes tempting.

Here, we describe the transcriptome of adult *C. elegans* GLR glia, uncovering similarities with both astrocytes and endothelial cells.

[1]Laboratory of Developmental Genetics, The Rockefeller University, 1230 York Avenue, New York, NY 10065, USA. [2]Research Bioinformatics, The Rockefeller University, 1230 York Avenue, New York, NY 10065, USA. [3]Present address: Alexion Pharmaceuticals, Boston, MA 02135, USA. ✉E-mail: shaham@rockefeller.edu

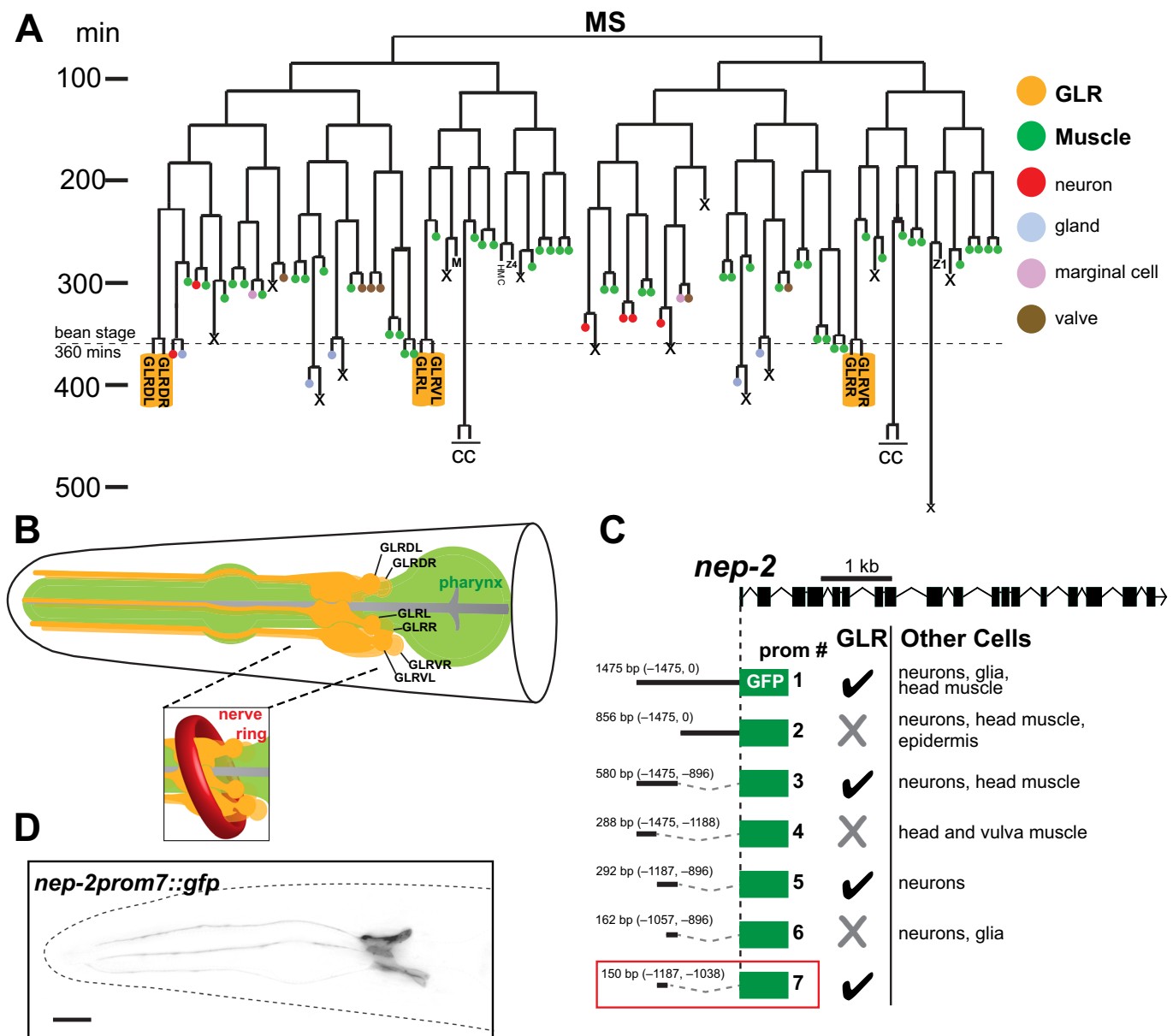

**Figure 1.   Generation of a GLR-specific driver to study the expression profile of the mesodermal GLR glia.**

(A) GLR glia (yellow boxes) derive from the lineage of the blast cell MS. This lineage produces mainly body wall muscle and pharyngeal muscle cells (green). GLR glia (yellow) are born at around the embryonic bean stage (360 min of embryonic development). The HMC cell and coelomocytes (CC) also derive from the MS lineage. Schematic adapted from (Sulston et al, 1983). (B) Schematic representation of the GLR glia (yellow). Pharynx is shown in green. The inset shows how *C. elegans* Nerve Ring (red) wraps around the sheet-like GLR glia processes. Schematic redrawn and modified from (Altun and Hall, 2016). (C) Cis-regulatory dissection analysis for the gene *nep-2* resulted in isolation of a GLR glia-specific driver, prom7 (red box). (D) Fluorescence image of an L4 *C. elegans* showing expression of *nep-2prom7::gfp* specifically in GLR glia. Anterior is left, dorsal is up, and scale bar is 10 μm. (E) Genes from three families (neurotransmitter receptors and transporters, potassium channels and extracellular matrix genes) are overrepresented among GLR-enriched genes.

We use the transcriptome to develop a molecular toolkit for labeling and manipulating GLR glia, which we use to identify and characterize two master regulators of GLR glia development. We show that LET-381, the *C. elegans* ortholog of the forkhead transcription factor FoxF, promotes GLR glia fate specification and maintenance and that UNC-30/Pitx2 transcription factor controls GLR glia morphology and represses the acquisition of an alternative mesodermal fate. Importantly, the expression of both transcription factors in a naive cell is sufficient to promote GLR glia gene expression. Through genetic studies, we order *let-381*, *unc-30*, and other genes into a pathway for GLR glia development. Finally, we show that *let-381* autoregulation-deficient mutants, as well as animals in which GLR glia are genetically ablated, exhibit specific defects in locomotory behavior and are hypersensitive to salt, suggesting important roles for these glia in coordinating neuronal activity.

## Results

### GLR glia gene expression reveals similarities with astrocytes and endothelial cells

Although GLR glia reporters have been previously described, none are exclusively expressed in these cells (Krause et al, 1994; Ringstad et al, 2009; Ringstad and Horvitz, 2008; Warren et al, 2001; Yamada et al, 2010). To identify drivers allowing specific genetic manipulation and marking of GLR glia, we performed promoter dissection studies of known GLR glia-expressed genes (Appendix Fig. S1A–E), isolating a 150 bp cis-regulatory sequence from the gene *nep-2* that, when fused to *gfp*, promotes expression only in GLR glia (Fig. 1C,D). Expression of this reporter is first detected in first-stage (L1) larvae and is maintained through adulthood.

To identify genes regulating GLR glia development and functions, we generated a stable transgenic *C. elegans* strain expressing nuclear YFP using the *nep-2* regulatory sequence (Appendix Fig. S1F). *nep-2prom7::nls::yfp* expressing cells were isolated from dissociated L4 larvae using fluorescence-activated cell sorting (FACS), and lysed to isolate mRNA. Following mRNA amplification and RNA-seq, we compared transcript abundances between GLR glia (YFP-positive) and all other cells (YFP-negative). We identified 886 genes with enriched GLR glia expression ($P < 0.05$, $\log_2$-fold enrichment >1, Dataset EV1) out of 13,794 genes with any GLR glia expression (>50 reads, Dataset EV1; Appendix Fig. S1G). To validate this list, we confirmed the expression of 39 GLR glia-enriched genes using transgenic and endogenous *gfp* reporters (green/blue highlights in Dataset EV1). In addition, all previously known GLR glia genes show strong enrichment in our analysis.

Using PANTHER gene ontology over-representation analysis (Mi et al, 2019; Thomas et al, 2022), we find that genes involved in synaptic transmission, including neurotransmitter receptors and transporters, potassium channels, and genes encoding extracellular matrix proteins, are overrepresented among GLR glia-enriched genes (Fig. 1E). Genes from these transcript classes as well as other GLR-enriched genes are also overrepresented in murine astrocytes (e.g., *snf-11/Gat1, gbb-1/Gabbr1, gbb-2/Gabbr2, glt-1/Glt1, ensh-1/Tnc, pll-1/Plcd4*) (Batiuk et al, 2020; Yang and Jackson, 2019; Zhang et al, 2014), suggesting similarities between GLR glia and astrocyte transcriptomes. Genes encoding ion, amino acid, and neurotransmitter transporters, as well as extracellular matrix proteins are enriched also in endothelial cells of the blood–brain barrier (Munji et al, 2019), and other GLR-enriched genes, including *let-381/Foxf, dep-1/Ptprb, tag-68/Smad6, T16A9.4/Ece1, gei-1/Dlc-1, slcf-2/Slc2a1, unc-115/Ablim1, mrp-2/Abcc6*, are also enriched in endothelial cells of the central nervous system (Batiuk et al, 2020; Munji et al, 2019; Zhang et al, 2014). In vertebrates, astrocyte endfeet are found in proximity to endothelial blood vessels, and in *C. elegans*, GLR glia separate the circulatory cavity from the nerve ring. It is intriguing to speculate that to conserve cell numbers, *C. elegans* may have merged astrocytic and endothelial functions into the GLR glia cell type. Such functional and anatomic compression has also been observed in the *C. elegans* motor circuit (Gao et al, 2018).

### *let-381/FoxF* is required early to specify GLR glia fate

To understand how the unusual fate merger of GLR glia arises, we sought to identify transcription factors that control GLR glia fate specification and differentiation (Table EV1). Transcripts encoding LET-381, the sole *C. elegans* ortholog of the Forkhead domain transcription factor FOXF (Amin et al, 2010), are highly enriched in GLR glia. Recent studies suggest that FoxF genes are expressed in phagocytic glia and independently in endothelial mural cells (Reyahi et al, 2015; Scimone et al, 2018), raising the possibility that LET-381 could govern a combined glia/endothelial gene expression pattern in GLR glia. To follow *let-381* expression in developing animals, we used CRISPR/Cas9 to insert *gfp* coding sequences into the endogenous *let-381* locus (Fig. 2A). Transgenic homozygotes display nuclear GFP fluorescence in likely GLR glia precursors (pre-bean; Fig. 2B), as previously identified by lineaging of a *let-381* transcriptional reporter (Murray et al, 2012), and in GLR glia until adulthood (Fig. 2B). Expression is also detected in the head mesodermal cell (HMC) and in coelomocytes (Fig. 2B; Appendix Fig. S2A), cell types also generated by the MS lineage (Fig. 1A). Animals carrying the *gfp* reporter transgene do not exhibit the lethality associated with loss of *let-381* (see below), suggesting that *let-381* gene function is retained.

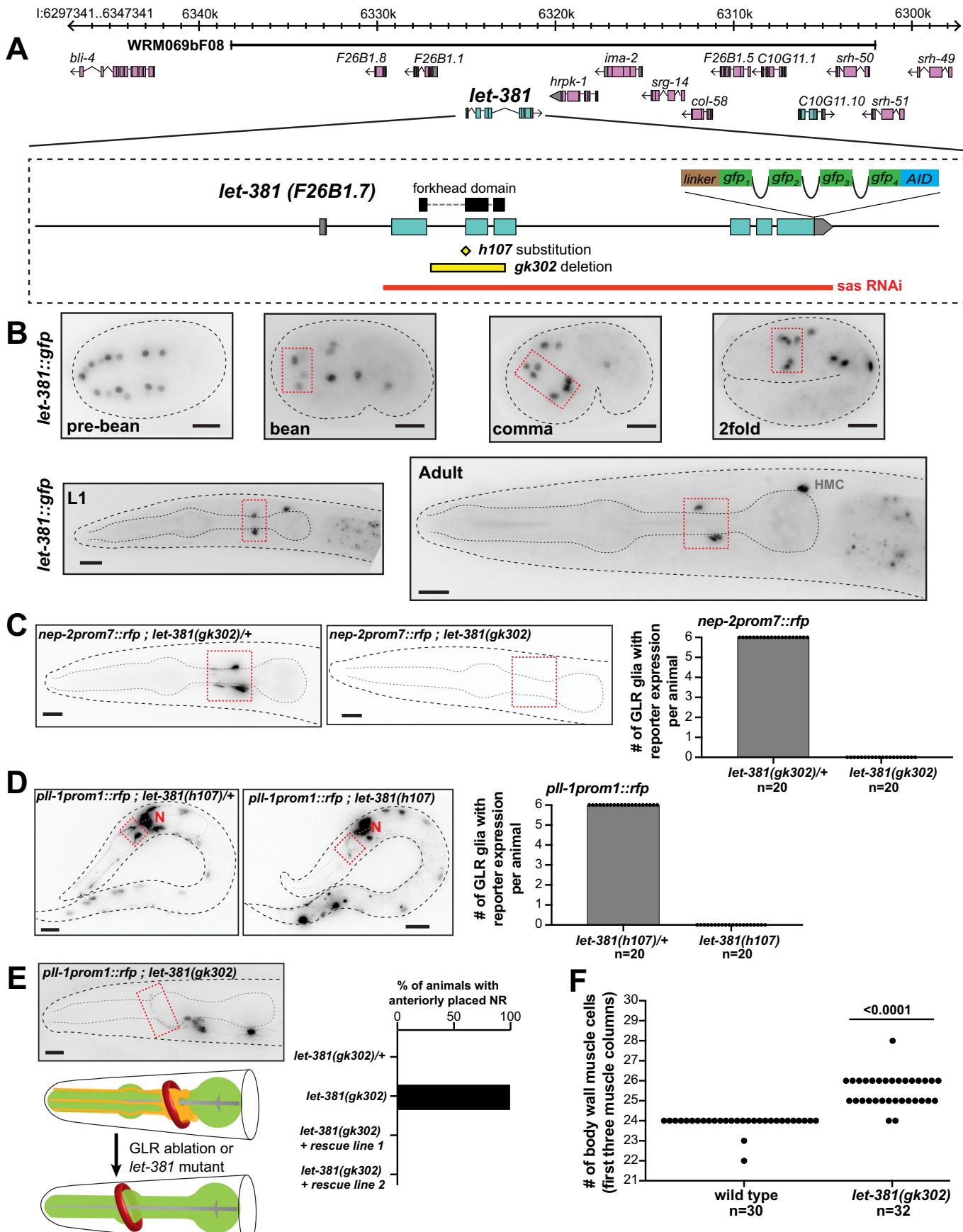

◄ **Figure 2. *let-381/FoxF* is required for GLR glia fate specification.**

(A) *let-381* genomic locus showing mutant alleles, reporters, fosmid genomic clones and RNAi sequences used in this study. (B) Expression of the endogenous *let-381::gfp* reporter at different stages during development. Dashed red boxes outline expression in GLR glia. (C) Absence of *nep-2prom7::rfp* expression in GLR glia (dashed red box) in homozygous *let-381(gk302)* mutants (right) as opposed to heterozygous animals (left). Quantification (number of GLR glia with *nep-2prom7::rfp* expression) is shown in the bar graph on the right. (D) Absence of *pll-1prom1::rfp* expression in GLR glia (dashed red box) in homozygous *let-381(h107)* mutants (right) as opposed to heterozygous animals (left). Quantification (number of GLR glia with *pll-1prom1::rfp* expression) is shown in the bar graph. Red "N" denotes *pll-1prom1::rfp* expression in neurons. (E) Similar to GLR glia-ablated animals (schematic), *let-381(gk302)* homozygous mutants lacking GLR glia exhibit anteriorly displaced nerve ring (NR). Dashed red box outlines a neuronal axon of the NR. Red circle in schematic indicates the Nerve Ring and pharynx is shown in green. Quantification is shown in the bar graph. Transgenic animals carrying the fosmid genomic clone WRM069bF08 with wild-type *let-381* (rescue lines 1 and 2) display normal NR position. (F) Number of body wall muscle cells in the first three muscle columns of head and neck in wild type and *let-381(gk302)* mutants. Data information: unpaired *t* test was used for statistical analysis in (F). Anterior is left, dorsal is up and scale bars are 10 μm for all animal images. Source data are available online for this figure.

To determine whether *let-381* promotes GLR glia fate specification, we introduced transgenic and endogenous GLR glia reporters into animals homozygous for the previously identified *let-381* null alleles *gk302* and *h107*. *gk302* contains a deletion spanning LET-381 DNA binding domain encoding sequences; and *h107* is a G to A substitution at a splice acceptor site (Fig. 2A). Animals homozygous for either allele undergo late-embryonic/early-larval arrest, with a few *gk302* animals surviving to become sterile adults (sterility may reflect improper sex muscle specification in *let-381* mutants (Amin et al, 2010)). Homozygous mutant *gk302* or *h107* animals fail to express five different GLR glia reporters that we tested (*nep-2prom7::gfp*, *pll-1prom1::rfp*, *gly-18prom::gfp*, *hlh-1::gfp*, *unc-30::gfp*) (Fig. 2C,D; Appendix Fig. S2B), suggesting that GLR glia are not generated in these mutants. Consistent with this, laser ablation of GLR glia precursors was previously shown to cause anterior displacement of the nerve ring (Shah et al, 2017) and we find a similar defect in *let-381* mutants (Fig. 2E). In addition, the GLR glia sister lineage produces head body wall muscle cells (Fig. 1A), and we observe extra muscle cells in the heads of *let-381(gk302)* mutants (Fig. 2F). All *let-381(gk302)* mutant defects are rescued by a transgene containing the wild-type *let-381* locus (*let-381* fosmid WRM069bF08; Fig. 2A,E; Appendix Fig. S2C,D). Taken together, our results suggest that LET-381 is required for the specification of GLR glia, and in its absence, some GLR lineages adopt sister muscle lineage fates instead.

## *let-381/FoxF* is continuously and cell-autonomously required to maintain GLR glia gene expression

Although LET-381 is required early to specify GLR glia, its expression during larval development and in adults suggests it may have additional later roles. To test this idea, we knocked down *let-381* in GLR glia of *let-381::gfp* animals by RNAi, using transgenic constructs co-transcribing sense and antisense *let-381* sequences (Esposito et al, 2007) from the postembryonic *nep-2prom7* promoter. These animals, also homozygous for the RNAi-sensitizing allele *eri-1(mg366)*, downregulate GFP expression specifically in GLR glia, but not in other *let-381* expressing cells, confirming RNAi specificity and efficacy (Fig. 3A). Importantly, *let-381* RNAi transgenes downregulate expression of GLR glia reporters for *nep-2*/neprilysin, *hlh-1*/MyoD/Myf, *snf-11*/GAT GABA transporter, *unc-46*/LAMP-like, *gly-18*/N-acetylglucosaminyl transferase, and *lgc-55*/tyramine receptor (Fig. 3B–G). Expression of *unc-30*/Pitx2 is not affected by *let-381* RNAi (see below), allowing us to visualize the cells and determine that they are still generated. Indeed, *let-381* RNAi animals neither display an

anteriorly displaced nerve ring nor have extra head muscle cells (Appendix Fig. S3A,B). Thus, LET-381 is required post-embryonically to maintain GLR glia gene expression, and this function is distinct from its role in generating GLR glia.

To probe dynamic functions of *let-381*, we used CRISPR/Cas9 to insert sequences encoding an auxin-inducible degron (AID) (Zhang et al, 2015) into the *let-381* genomic locus (Fig. 2A). In the presence of GLR glia-expressed TIR1 and a synthetic auxin analog (K-NAA) (Martinez et al, 2020), AID-tagged LET-381 protein is degraded within 2 hours specifically in GLR glia (Fig. 4A,C, upper left panel; Appendix Fig. S4). A three-day exposure to K-NAA starting either at the L1 or late-L4/young-adult stages downregulates *nep-2prom7::rfp* reporter expression (Fig. 4B–F). Thus, *let-381* functions cell-autonomously and is continuously required to maintain GLR glia gene expression, even in adults.

## LET-381 binding motifs are required and sufficient for GLR glia gene expression

The expression of LET-381 in GLR glia throughout its life suggests it may function as a terminal selector, directly co-regulating expression of terminal differentiation gene batteries via a shared cis-regulatory motif (Hobert, 2011). To test this idea, we used sequences identified in our promoter dissection studies, ranging in size from 125 to 1022 bp, as input for the motif discovery tool MEME (Bailey and Elkan, 1994). This analysis identified a TGTTTA(C/T/G)A sequence common to all sequence inputs (Fig. EV1A). Remarkably, this sequence is highly similar to a previously identified FoxF binding sequence in mice (Peterson et al, 1997) and to a *C. elegans* LET-381 binding sequence identified through protein binding microarrays (Narasimhan et al, 2015) (Fig. EV1B,C). Notably, we find this motif in regulatory regions of all genes whose expression in GLR glia is downregulated by *let-381* knockdown (Fig. EV1D). We refer to the TGTTTA(C/T/G)A sequence as the *let-381* motif.

To assess how the motif affects gene expression, we used CRISPR/Cas9 to mutate it in different genomic locations (Fig. EV1F). As shown in Fig. 5, mutating *let-381* motifs in upstream regulatory regions of the *nep-2* and *pll-1* genes, fused endogenously to *gfp*, significantly reduces *gfp* expression (Fig. 5A,B). Animals homozygous for an endogenous *inx-18::gfp* insertion allele we generated localize GFP in bright puncta marking the gap junctions between the GLR glia and the RME motoneurons. Mutagenesis of the *inx-18 let-381* motif eliminates these bright puncta (Fig. 5C). Finally, disrupting either of two motifs in the gene *hlh-1* only slightly reduces endogenous *hlh-1::gfp* expression;

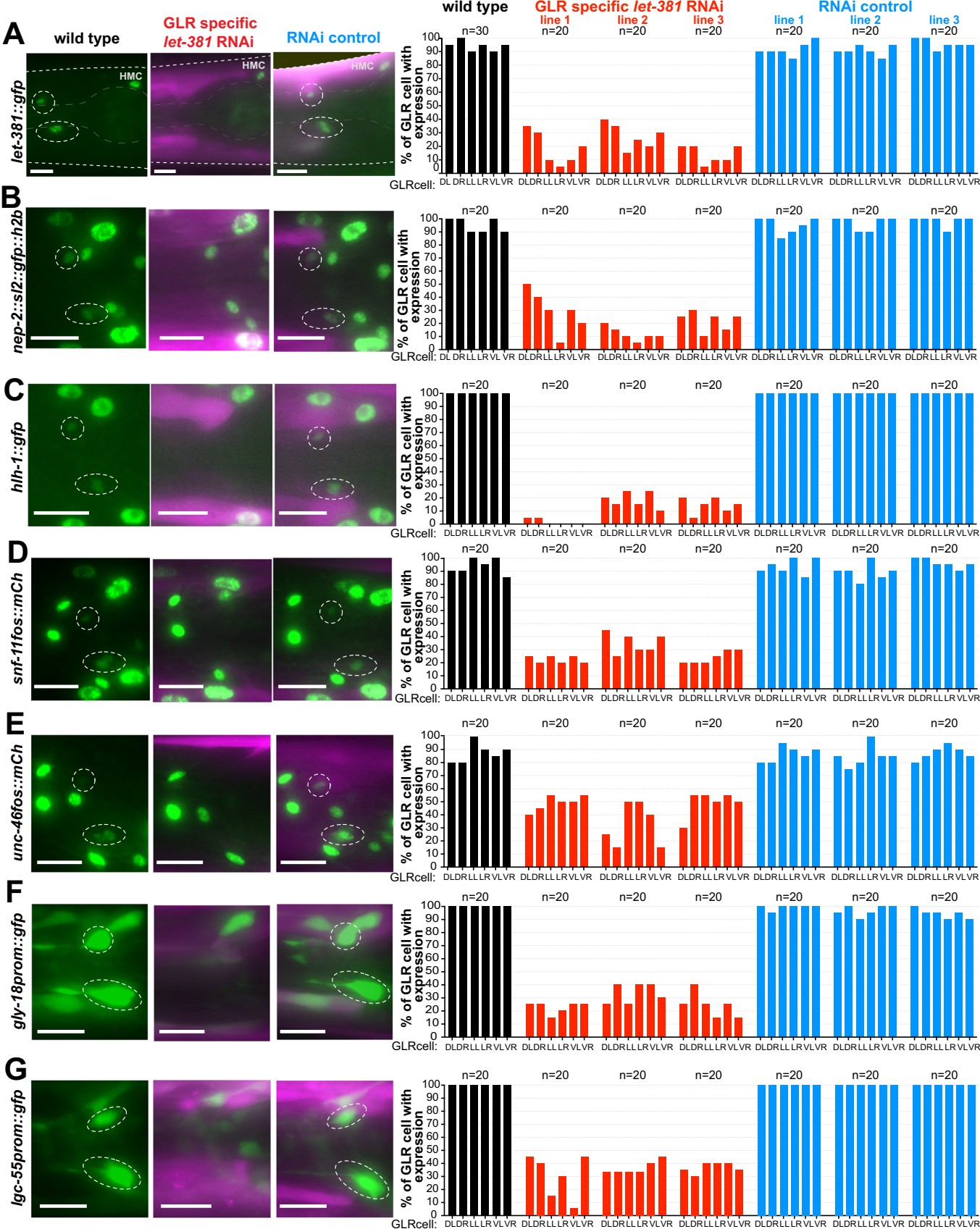

**Figure 3. Postembryonic *let-381* knockdown affects GLR glia gene expression.**

(A–G) *gfp* or *mCherry*-based reporter expression (green) in GLR glia of endogenously tagged (**A**) *let-381*, (**B**) *nep-2*, and (**C**) *hlh-1* and transgenic (**D**) *snf-11* (**E**) *unc-46*, (**F**) *gly-18*, and (**G**) *lgc-55* in wild-type (left column), GLR-specific postembryonic *let-381* RNAi (middle column) and RNAi control animals (right column). Fluorescence images of L4 animals are shown. GFP expression in GLR glia (dashed white circles) is downregulated in the *let-381* RNAi animals but not affected in RNAi control. RNAi and control lines carry a co-injection marker expressed in body wall muscle (magenta). Quantification is shown in bar graphs on the right. Each bar represents % of expression in each of the six GLR glia (DL, DR, LL, LR, VL, VR) in the three different backgrounds (wild type = black, GLR-specific RNAi = red, RNAi control = blue). Three independent extrachromosomal lines were scored for the RNAi and RNAi controls. Data information: anterior is left, dorsal is up, and scale bars are 10 μm for all animal images. Source data are available online for this figure.

however, disrupting both together completely abolishes expression (Fig. 5D), suggesting that in some contexts, LET-381 binds multiple sites in the same gene.

To further test the idea that *let-381* is a terminal selector gene, we identified motifs in 15 GLR glia-enriched genes whose expression was not previously verified. We then generated animals transgenic for sequences surrounding *let-381* motifs from each gene fused to *gfp* coding sequences (Table EV2). We found that *let-381* motif-containing regions (ranging in size from 149 to 254 bp) from 14/19 genes are sufficient to drive GFP expression in GLR glia. Our experiments, therefore, support the idea that *let-381* is a terminal selector gene, controlling the coordinate expression of genes expressed in differentiated GLR glia.

## LET-381 regulates its own expression to maintain GLR glia identity

Given the requirement for *let-381* in maintaining GLR glia gene expression even in adults, we wondered how *let-381* expression itself is maintained. We identified a conserved *let-381* motif upstream of the *let-381* first exon (Fig. EV2A,B). We wondered whether, through positive feedback, this element could account for sustained *let-381* expression through adulthood, and used CRISPR/Cas9 to delete this motif. In mutant animals, *let-381(ns1026)*, endogenous *let-381::gfp* expression is observed in GLR glia of bean-stage embryos at levels similar to wild type. However, expression gradually wanes and is completely lost in L1 larva and older animals (Fig. 6A). Thus, *let-381* is required to maintain its own expression through an autoregulatory *let-381* motif. Expression of nine downstream GLR glia genes (but not *unc-30*, see below) is also gradually lost by the L1 stage (Figs. 6B–E and EV2C–G), further supporting the notion that LET-381 is required for GLR glia fate maintenance. Of note, *let-381* autoregulation mutants exhibit neither an anteriorly displaced nerve ring nor extra head muscles, consistent with the RNAi and AID knockdown results.

## *unc-30/Pitx2*, a *let-381/FoxF* target, controls GLR glia gene expression and shape

Our transcriptome studies revealed that transcripts encoding the transcription factor UNC-30/Pitx2 are also significantly enriched in GLR glia (Table EV1). UNC-30 was previously identified as a terminal selector of GABAergic identity in ventral cord neurons (Cinar et al, 2005; Eastman et al, 1999; Jin et al, 1994), and a recent study showed that GLR glia are GABA-positive by immunostaining and express the GABA-related genes *snf-11/GAT*, *gta-1/GABA-T*, and *unc-46/LAMP* (Gendrel et al, 2016). We found that animals carrying an endogenous *unc-30::gfp* reporter we generated (Fig. 7A) express GFP in GLR glia starting at the embryonic bean stage and

through to adulthood (Fig. 7B). We also observed *unc-30::gfp* expression in the ASG, AVJ, DD, VD, and PVP neuron classes, all derived from the AB lineage (Fig. 7B; Appendix Fig. S5A).

To determine whether *unc-30* controls GLR glia gene expression, we crossed gene reporters into *unc-30(e191)* mutants, harboring an early nonsense mutation (Fig. 7A). Endogenous *hlh-1::gfp* expression is completely abolished in these animals, and expression of endogenous *nep-2::gfp*, transgenic *nep-2prom7::gfp*, or *lgc-55prom::gfp* is also lost, but mainly in lateral and ventral GLR glia (Figs. 7C,D and EV3B–D). Expression of transgenic *gly-18prom::gfp*, *pll-1prom1::gfp* and *snf-11fosmid::SL2::mCherry:H2B* reporters is reduced but not abolished (Fig. EV3E–G). In most animals in which reporter expression is not extinguished, GLR glia anterior processes are shortened (Fig. 7C,E). Similar findings are observed with the *unc-30(ok613)* deletion mutant (Fig. 7A,C,E), and all GLR glia defects are rescued with transgenes carrying the wild-type *unc-30* locus (Fig. EV3A). The segregation of these unstable extrachromosomal rescuing transgenes to MS-lineage but not AB-lineage cells is sufficient to rescue GLR defects of *unc-30(e191)* mutants, suggesting that UNC-30 functions cell-autonomously in GLR glia (Appendix Fig. 5B–D). Thus, UNC-30 regulates GLR glia gene expression and also controls GLR glia morphology. Gene expression of dorsal GLR glia appears largely unaffected in *unc-30* mutants. This suggests that UNC-30 has a more restricted effect on GLR gene expression than LET-381, and that other, yet unidentified transcription factors, may act as LET-381 cofactors to control gene expression in the dorsal GLR.

How might *let-381* and *unc-30* interact? We found that the fifth intron of *unc-30* contains three conserved *let-381* motifs located within an 88 bp sequence (Fig. EV3H,I). Sequences derived from this intron are sufficient to promote GFP expression in GLR glia (Fig. EV3H). Furthermore, mutating the upstream *let-381* motif alone substantially reduces endogenous *unc-30::gfp* expression, and simultaneous mutation of the two upstream motifs, a mutation of the downstream motif alone, or a 169 bp deletion removing all three motifs all specifically abolish *unc-30::gfp* expression in GLR glia but not in other *unc-30* expressing cells (Figs. 7F,G and EV3I). Remarkably, the *unc-30(ns998)* 169 bp deletion allele, which abolishes *unc-30::gfp* expression only in GLR glia, phenocopies the effect of *unc-30* null alleles on GLR glia gene expression and anterior process length (Figs. 7D and EV3B), supporting the notion that *unc-30* functions cell-autonomously in GLR glia. We found no effects of *unc-30* loss on the expression of *let-381*; however, *let-381* expressing cells are often displaced along the dorsoventral and left-right axes in *unc-30* mutants (Fig. 8A,B). Together, these results support the conclusion that *unc-30* acts cell-autonomously to control GLR glia gene expression and anterior process length and that *let-381* controls *unc-30* expression in GLR glia. Previous studies suggest that UNC-30 may regulate its own

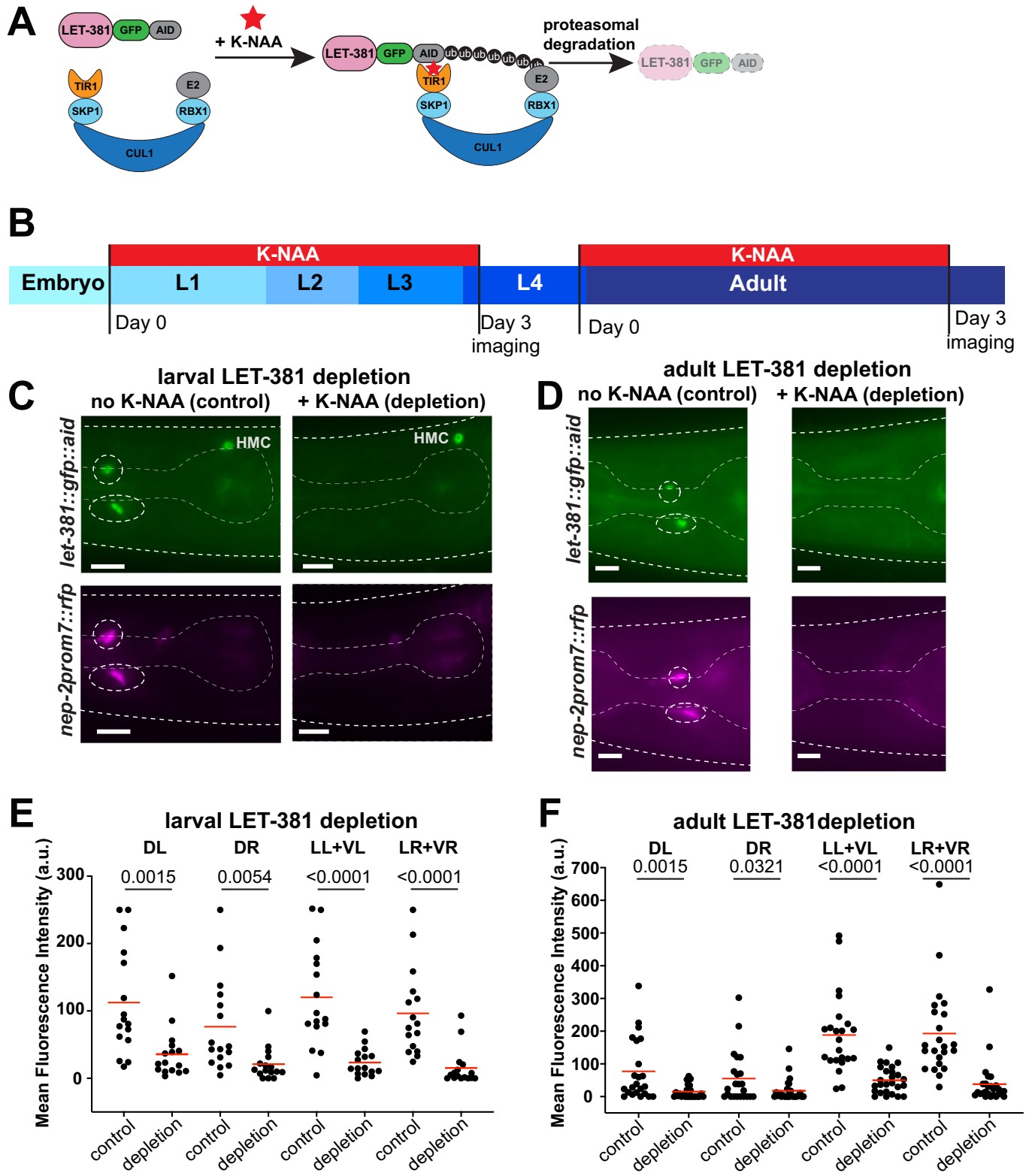

expression (Cinar et al, 2005; Hobert, 2011), suggesting that following initial LET-381 binding, *unc-30* may become independent of *let-381*. This notion is supported by our findings that *unc-30::gfp* expression in GLR glia is unaffected when *let-381* is knocked down post-embryonically (Fig. EV3J,K).

## *unc-30/Pitx2* represses expression of HMC genes in GLR glia

Wild-type GLR glia have small, sesame-seed-shaped nuclei. We noticed that in *unc-30* mutants, GLR glia nuclei are larger and

Figure 4. Acute larval and adult LET-381 depletion results in loss of GLR gene expression.

(A) Schematic representation of auxin-induced degradation (Zhang et al, 2015) of LET-381. TIR1 is transgenically provided and expressed specifically in GLR glia by the *nep-2prom7* promoter. (B) Timeline of the auxin-inducible depletion experiment. Synchronized populations of L1 or late-L4/Young-adult animals were placed on plates containing the auxin analog K-NAA and imaged after a three-day exposure to K-NAA. Fluorescence intensities of gene expression in GLR glia were compared to age-matched animals grown on control plates without K-NAA. (C, D) Fluorescence images showing the result of (C) larval and (D) adult depletion of LET-381 using auxin-inducible degradation. Animals grown on control plates without K-NAA (left panels) show expression of *let-381::gfp* (green) and *nep-2prom7::rfp* (magenta) in GLR glia (dashed white circles). Age-matched animals grown on K-NAA-containing plates (right panels) show depletion of endogenous *let-381::gfp* expression specifically in the GLR glia; as shown expression in HMC remains unaffected. As a result, expression of the GLR-specific *nep-2prom7::rfp* reporter is downregulated in GLR glia. (E, F) Quantification (mean fluorescence intensity in cell bodies) of (E) larval and (F) adult LET-381 depletion on expression of *nep-2prom7::rfp* reporter in each GLR glia. Cell bodies of Lateral and Ventral GLR glia are too close to be clearly distinguished, thus they were grouped (LL + VL, LR + VR) for quantification purposes for this experiment. Red lines in dot plots indicate averages. Data information: unpaired *t* test used for statistical analysis in (E, F). *n* = 16 for control and depletion in (E), *n* = 23 for control and *n* = 26 for depletion in (F). a.u. = arbitrary units. Anterior is left, dorsal is up and scale bars are 10 μm for all animal images. Source data are available online for this figure.

rounder, resembling the nucleus of the head mesodermal cell (HMC), another *let-381*-expressing cell (Fig. 8A). To determine whether *unc-30* loss results in GLR glia acquiring additional HMC characteristics, we generated a *nep-2prom7::tagrfp* GLR glia reporter strain also expressing *arg-1prom::gfp*, an HMC reporter. We found that in *unc-30(e191)* mutants, some GLR glia that lose RFP expression now ectopically express the HMC reporter. *unc-30(ns998)* mutants exhibit similar defects (Fig. 8C,D). Likewise, two additional HMC reporters, *glb-26prom::gfp* and *dmd-4::his24::m-Cherry*, are also mis-expressed in GLR glia of *unc-30* mutants (Fig. EV4A–C). While HMC-converted GLR glia retain *let-381::gfp* expression (Fig. EV4D), we never observe GLR glia expressing a mix of GLR glia-specific and HMC-specific reporters. Finally, we found that ectopic expression of the HMC reporter *arg-1prom::gfp* requires the DMRT transcription factor DMD-4 (Figs. 8D,E and EV4E), normally required for *arg-1prom::gfp* expression in the HMC (Bayer et al, 2020). We conclude that UNC-30 acts in GLR glia to repress *dmd-4*-dependent HMC-specific gene expression. This model succinctly explains the differential effect of *unc-30* loss on the various reporters we tested (Fig. EV4B–G): *snf-11*, *gly-18* and *pll-1* reporters are normally expressed in both GLR glia and HMC and are therefore less affected by an *unc-30* mutation than *nep-2*, *lgc-55*, or *hlh-1*, which are expressed in GLR glia but not in HMC.

Does *unc-30* control of GLR glia-specific gene expression require inhibition of HMC gene expression? We found that even though the total number of cells expressing *let-381::gfp* is unaltered between wild type and *unc-30* mutants (6 total cells—excluding HMC; Fig. 8B), the number of cells expressing either a GLR glia or an ectopic HMC reporter rarely adds up to 6 (Fig. EV4F), indicating that there are GLR glia that lose GLR glia gene expression without acquiring HMC fate. Thus, it is likely that *unc-30* mediates GLR glia-specific gene expression independently of HMC gene expression suppression.

### *let-381* and *unc-30* are together sufficient to induce GLR glia gene expression

The broader effect of *let-381* loss on GLR glia gene expression suggests that expression of this gene in naive cells should drive the GLR glia expression program in these cells. Surprisingly, we found that this is not the case (Fig. 8F). Indeed, broad inducible misexpression of *let-381* cDNA using a heat-shock promoter results in minimal misexpression of the *nep-2prom7::tagrfp* GLR glia reporter. We found a similar result using *unc-30* cDNA alone. However, misexpression of both *let-381* and *unc-30* results in

highly penetrant misexpression of *nep-2prom7::tagrfp* in many cells, including body wall muscle, pharyngeal muscle, stomato-intestinal muscle and ventral nerve cord motoneurons (Fig. 8F,G). Thus, *let-381* is not sufficient to induce GLR glia fate and must cooperate with *unc-30*.

### GLR glia-defective animals display locomotion abnormalities

Our development of tools to manipulate gene expression and function specifically in GLR glia allowed us to interrogate the functions of these cells. To do so, we generated a *nep-2prom7::egl-1* transgenic line driving postembryonic expression of EGL-1, a pro-apoptotic caspase activator, only in GLR glia. We observed GLR glia loss, rescued by the *ced-3(n717)* caspase mutation, confirming that GLR glia in this strain die by apoptosis (Fig. 9A–C). We next recorded movies and analyzed the locomotion of GLR glia-ablated animals freely moving on an agar plate without food (Katz et al, 2018; Katz et al, 2019). Remarkably, we found that while GLR glia-ablated animals can move, they exhibit severe defects in all locomotion parameters we analyzed, including reduced locomotion rates, increased turning frequency, increased reversal rates, and increased pausing (Fig. 9D–K; Movies EV1 and EV2). These defects are similar to those observed following ablation of CEPsh glia, *C. elegans* astrocytes that line the outer aspect of the nerve ring and regulate synaptic function (Katz et al, 2018; Katz et al, 2019). In addition, while wild-type animals exposed to high-salt concentrations are initially paralyzed and then recover, GLR glia-ablated animals paralyze more quickly and recover at a much slower rate (Fig. 9L), suggesting that GLR glia may play an important role in the control of solute permeability and ionic balance in the nerve ring. Thus, GLR glia are important for coordinated locomotion and nerve ring function.

We noted that, like animals whose GLR glia progenitors are laser ablated during embryogenesis (Shah et al, 2017), *nep-2prom7::egl-1* ablated animals have anteriorly displaced and sometimes defasci-culated nerve rings (Fig. EV5A). To determine whether the abnormal locomotion we observed is a consequence of these structural defects, we examined the behavior of axon-guidance mutant strains that also possess anteriorly displaced nerve rings (Kennerdell et al, 2009; Zallen et al, 1999). Indeed, these mutants also exhibit defects in all locomotion parameters tested. However, the magnitude of these defects is not always the same as in GLR glia-ablated animals (Fig. EV5B). Thus, while some of the locomotion defects may be attributed to structural defects in the nerve ring, it appears that animals lacking GLR glia may be

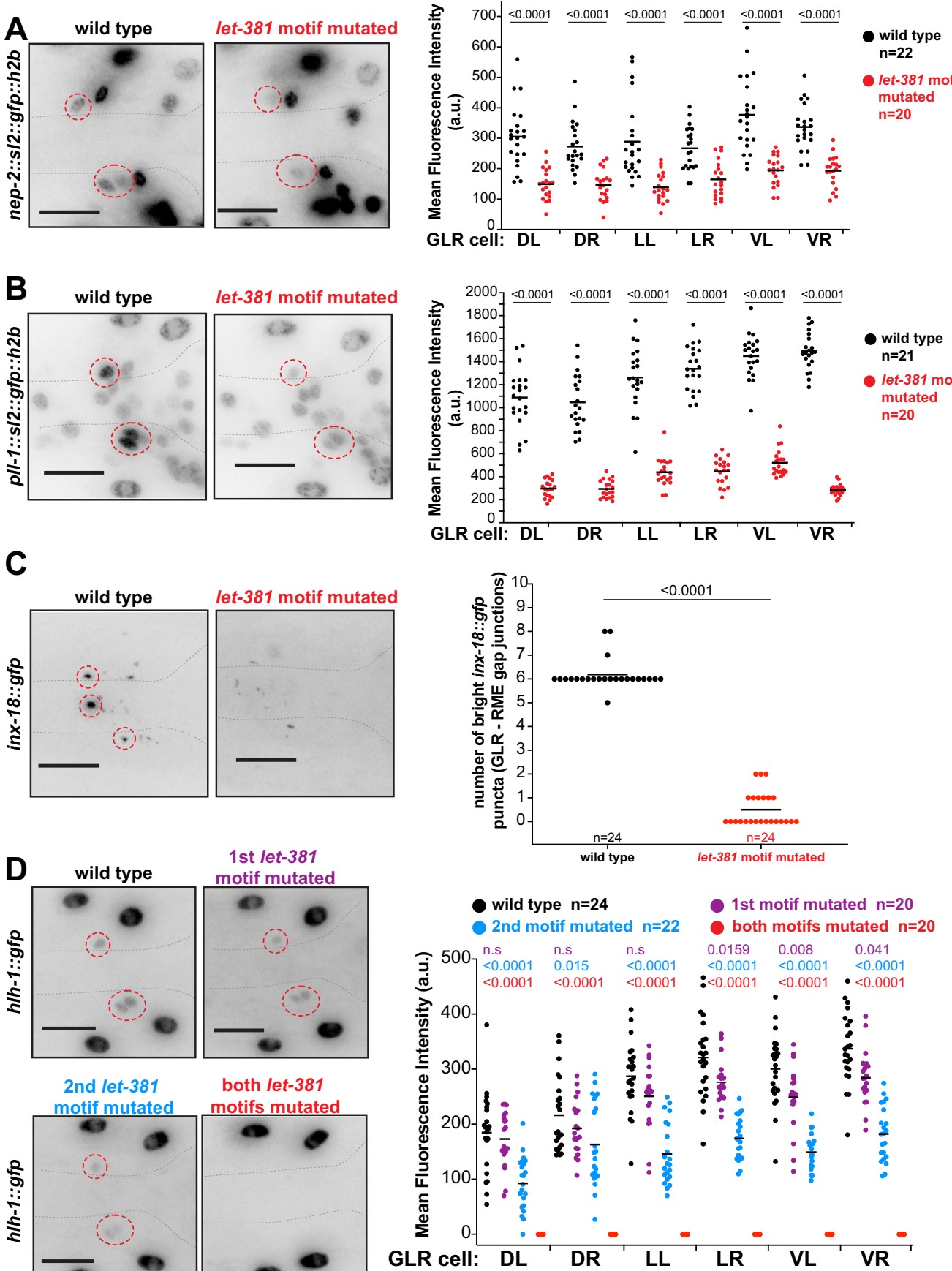

◄

**Figure 5.** *let-381* motifs are required for endogenous gene expression in GLR glia.

(A–D) Endogenous expression of (A) *nep-2*, (B) *pll-1*, (C) *hlh-1*, and (D) *inx-18* in wild-type and *let-381* motif-mutated animals (details on endogenous gfp reporters and molecular identity of *let-381* motif mutations are shown in Fig. EV1F). Animal images are on the left. Dashed circles outline expression in GLR glia. Quantifications are shown in the dot plots on the right. Data information: Mean fluorescence expression intensity for each GLR glia cell for (A, B, D) and number of bright gap-junction puncta for (C) is compared between the wild-type and *let-381*-motif-mutated backgrounds. Black lines indicate averages. Unpaired *t* test used for statistical analysis. a.u. = arbitrary units. Anterior is left, dorsal is up, and scale bars are 10 μm for all animal images. Source data are available online for this figure.

additionally compromised. To uncouple the physical positioning of the nerve ring from other GLR glia functions in *C. elegans* locomotion, we examined movement of animals carrying the *let-381* autoregulatory motif mutation, which does not result in nerve ring displacement. This strain, *let-381(ns1026)*, was generated from the *let-381(ns995)* strain, containing an insertion of *gfp* into the *let-381* locus. As in GLR glia-ablated animals, we find specific defects in *let-381(ns1026)* mutants: animals change direction more frequently than *let-381(ns995)* controls (Fig. 9D), display a higher frequency of reversals (Fig. 9E), and tend to pause more often (Fig. 9F). Other behaviors defective in ablated animals are unaltered (Fig. 9G–K). Auxin-dependent LET-381::AID downregulation late in development (24 hour auxin exposure of L4 animals) does not disturb nerve ring positioning and affects locomotion to the same extent as the *let-381(ns1026)* autoregulatory mutation (Fig. EV5C). Unlike GLR ablation, neither *let-381* autoregulatory mutant nor the *let-381*::AID knockdown result in hypersensitivity to high salt. We can only speculate that as opposed to GLR ablation, the GLR glia processes are still physically present in these mutants, and may therefore constitute enough of a physical barrier around the nerve ring, to protect it from the effects of sudden salt concentration shifts. Taken together, our observations support the conclusion that GLR glia regulate *C. elegans* locomotory behavior not only by ensuring nerve ring positioning but perhaps in non-structural ways as well, revealing a previously uncharacterized function for these cells.

## Discussion

We describe here a gene regulatory network for the specification and maintenance of *C. elegans* GLR glia (Fig. 10A). Early in development, *let-381/FoxF* is required to specify GLR glia fate and to suppress sister-lineage body wall-muscle gene expression. It does so, in part, by promoting the expression of dozens of GLR glia-enriched genes, all of which possess cis-regulatory LET-381 binding motifs. Such a motif upstream of *let-381* ensures that once turned on, the gene remains continuously expressed, as are its targets. Among these targets is *unc-30/Pitx2*, required for the expression of some *let-381*-dependent genes and for generating GLR glia anterior processes of appropriate length. *unc-30* also independently prevents expression in GLR glia of genes and traits of the HMC, a non-contractile MS-lineage-derived cell (Choi et al, 2023). Together, LET-381 and UNC-30 can bestow GLR glia gene expression onto naive cells that do not normally express these transcription regulators.

While transcriptional control of glial cell fate is generally not well understood, neuronal fate specification by terminal selector transcription factors has been studied extensively in *C. elegans*, *Drosophila*, chordates, and mice (Hobert and Kratsios, 2019). These master regulators, often acting with a specific cofactor (Hobert,

2011; Hobert and Kratsios, 2019), establish and maintain neuron identity by co-regulating a wide variety of neuronal features, including gene expression and synaptic connectivity. Terminal selectors also ensure neuron identity by repressing alternative neuronal fates (Feng et al, 2020; Remesal et al, 2020). Here we show that similar fate control is exercised in *C. elegans* GLR glia by *let-381/FoxF* and its cofactor *unc-30/Pitx2*. It is therefore plausible that such mechanisms also specify the fates of glial cells in other settings, and may account for regional differences between glia in the vertebrate brain.

The formation of intestinal muscle from visceral mesoderm using FoxF transcription factors is conserved among *Drosophila* (Zaffran et al, 2001), planaria (Scimone et al, 2018), *Xenopus* (Mahlapuu et al, 2001) and the mouse (Ormestad et al, 2006; Tseng et al, 2004). In *C. elegans*, *let-381/FoxF* is not expressed in the 4 intestine-associated muscles. Nonetheless, it is expressed in GLR glia, the head mesodermal cell (HMC), and the endocytic coelomocytes, all derived from the mesoderm-like lineage of the blast cell MS. Similar to our findings in GLR glia, *let-381/FoxF* functions with *ceh-34/Six2* in coelomocytes (Amin et al, 2010). Intriguingly, in planaria, *FoxF1* promotes specification of phagocytic cells, including phagocytic glia (Scimone et al, 2018). These findings suggest that FoxF transcription factors are key cell-fate regulators of non-contractile mesodermally derived cells, including mesodermal glia. Indeed, single-cell sequencing of mouse CNS cells reveals *FoxF1/2* expression in blood vessel endothelial and mural cells (Hupe et al, 2017; Saunders et al, 2018; Vanlandewijck et al, 2018; Zhang et al, 2014), with *FoxF2* driving pericyte differentiation and maintenance of the blood–brain barrier (Reyahi et al, 2015). While *FoxF* genes are expressed only at low levels in microglia (Vanlandewijck et al, 2018), it remains possible that these genes direct early microglia differentiation.

Intriguingly, cerebellar astrocytes, which often ensheath blood vessels, also express *FoxF1* (Kalinichenko et al, 2003), raising the possibility that *FoxF* genes play a broader role in specifying the entire neurovascular unit. Supporting this idea, we show here that *C. elegans* GLR glia merge astrocyte and mural cell molecular and anatomical characteristics (Fig. 10B,C). In phylogenetically older species, such as sturgeons (subclass: *Chondrostei*), astrocytes are thought to be the main components of the blood–brain barrier (Bundgaard and Abbott, 2008). Might GLR glia, by analogy then, form a barrier that isolates the *C. elegans* nerve ring from the pseudocoelom? Unlike endothelial cells, GLR glia do not adhere to each other to form a tight seal, and gaps between the cells are evident on electron micrographs (White et al, 1986). Furthermore, neuronal cell bodies, which reside outside the nerve ring, are not ensheathed and could have direct access to materials within the pseudocoelom. Nonetheless, roles for GLR glia in blocking diffusion of synaptically released factors into the pseudocoelom, which can then have systemic effects, are plausible, as are more general roles in regulating the extracellular environment of the

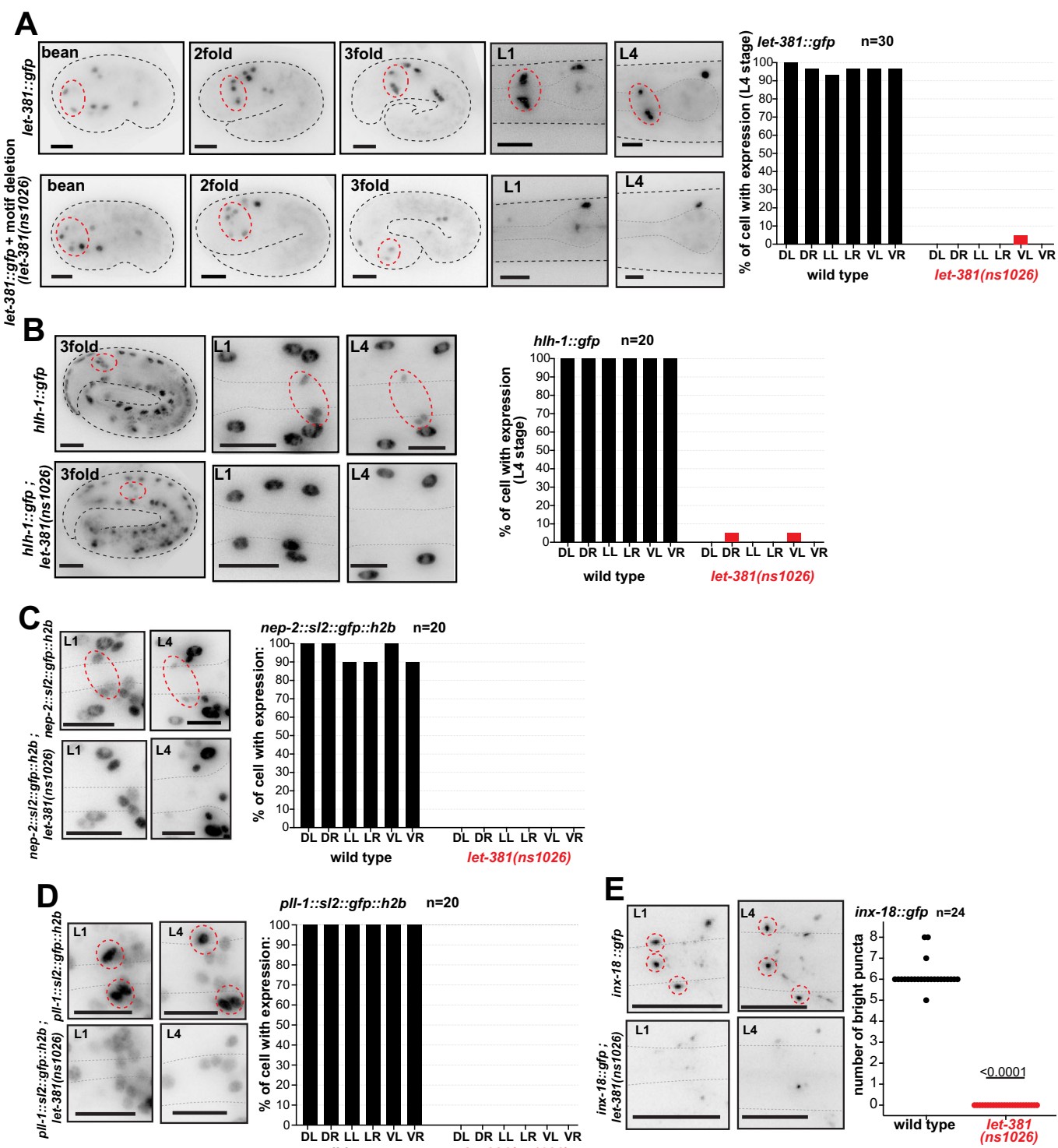

**Figure 6. *let-381* positively regulates its own expression.**

(A) Endogenous *let-381::gfp* expression in different development stages in two different backgrounds: wild type (top), *let-381* autoregulatory motif deletion (bottom). Animal images are shown on the left and quantification of expression in the L4 stage is shown on the right: percentage of each GLR cell expressing *let-381::gfp* in wild-type and autoregulatory motif deletion backgrounds. Red dashed circles outline GLR glia. (B–E) Effect of autoregulatory motif deletion on endogenous (B) *hlh-1*, (C) *nep-2*, (D) *pll-1*, and (E) *inx-18 gfp* reporter expression in GLR glia at different developmental stages. Quantification is shown on the right of each animal image panel for L4 animals. Red dashed circles outline GLR glia. Data information: unpaired *t* test used for statistical analysis in (E). Anterior is left, dorsal is up, and scale bars are 10 μm for all animal images. Source data are available online for this figure.

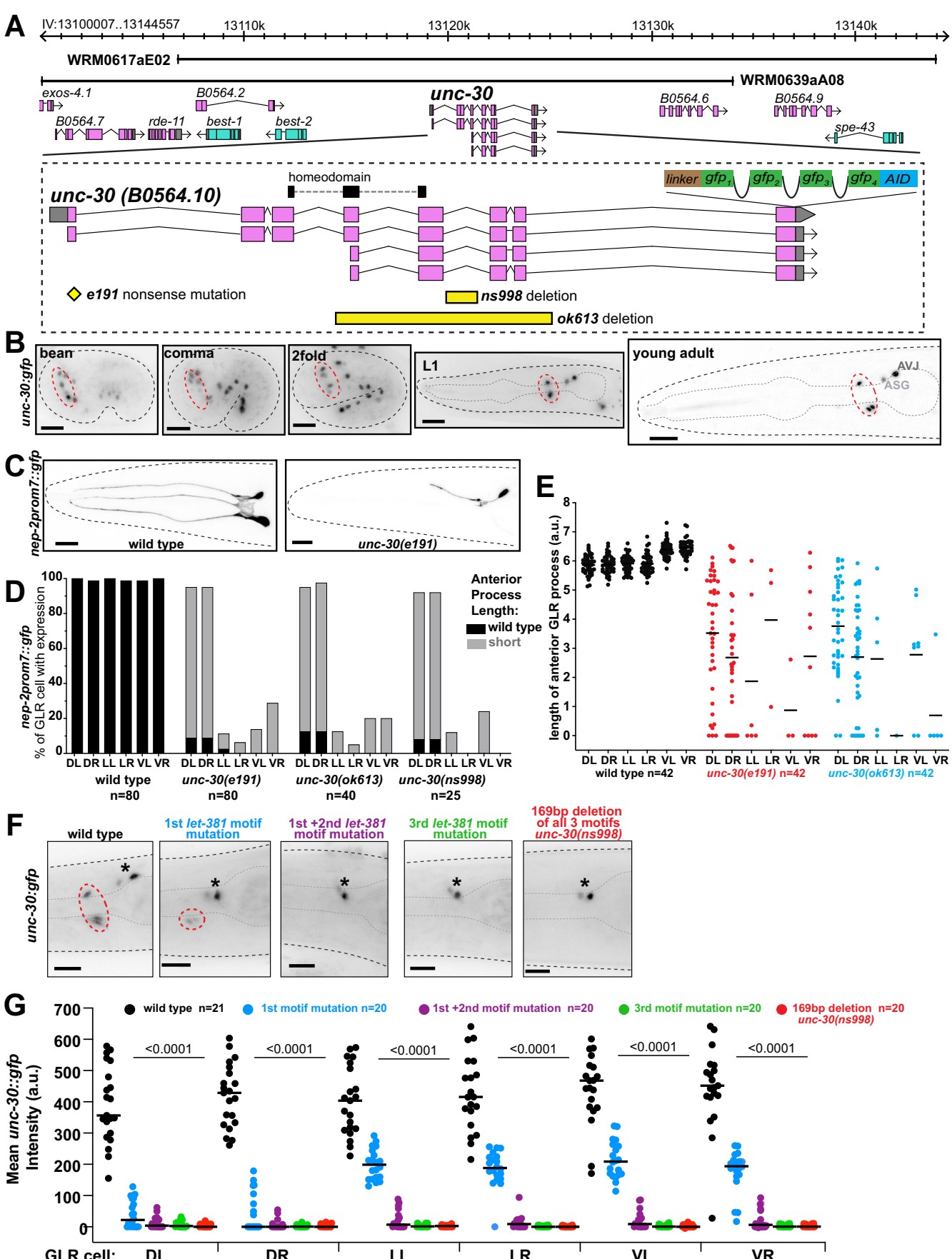

**Figure 7. *unc-30* acts downstream of *let-381* to control GLR glia gene expression and the length of GLR anterior process.**

(A) *unc-30* genomic locus showing mutant alleles, reporters, fosmid genomic clones used in this study. (B) Expression of the endogenous *unc-30::gfp* reporter at different stages during development. Dashed red circles outline GLR glia. (C) *nep-2prom7::gfp* expression in a wild-type L4 (left) and a *unc-30(e191)* null mutant L4 animal (right). GFP expression is lost in the lateral and ventral GLR glia in *unc-30(e191)*. The anterior process of the dorsal GLR glia still expressing GFP is shorter than that of a wild-type animal. (D) Quantification of percentage of each GLR glia cell with *nep-2prom7::gfp* expression (L4 stage) for different *unc-30* mutant backgrounds. (E) Quantification of the length of GLR anterior processes (L4 stage) for different *unc-30* mutant backgrounds. (F) Images of L4 animals showing differences in endogenous *unc-30::gfp* expression upon mutation of the *let-381* motifs present in the fifth intron of *unc-30*. Dashed red circles outline GLR glia and black asterisks denote expression in the ASG and AVJ head neurons. (G) Quantification of *unc-30::gfp* fluorescence intensity for each GLR glia cell; black lines in bar graphs indicate averages. Data information: unpaired *t* test used for statistical analysis in (G). a.u. = arbitrary units. Anterior is left, dorsal is up, and scale bars are 10 μm for all animal images. Source data are available online for this figure.

nerve ring. Indeed, our behavior studies of GLR glia-ablated animals and GLR glia mutants reveal disruptions in the coordination of locomotion, suggesting effects on neuronal signaling within the nerve ring. GLR glia are electrically coupled with the RME head motoneurons and head muscle and thus motor defects of GLR mutants could be due to disruption of synchronized activity of this gap-junction network. Alternatively, GLR glia mediated changes in neurotransmitter or neuropeptide signaling of an underlying motor network could account for the observed defects; indeed, some GABA, glutamate, acetylcholine, tyramine and putative neuropeptide receptors are enriched in GLR glia. GLR glia could also be important for regulation of potassium homeostasis in the extrasynaptic space, as supported by the over-representation of TWIK potassium channels among GLR-enriched genes, therefore having a broader role on neuronal excitability and synchronization. While nerve ring positioning is not affected in *let-381(ns1026)* mutants, microscopic structural defects may exist in individual neurons. Dissecting the effects of GLR glia mutants on neuronal development and activity would therefore be crucial to understand GLR glia roles in regulation of motor behavior. The similarities in locomotory defects between CEPsh astrocyte-ablated animals and GLR glia-ablated animals suggest that both glial types affect a similar set of underlying processes. CEPsh glia wrap around the outside aspect of the nerve ring, are not in direct contact with the pseudocoelom, and are not enriched for endothelial genes, further supporting the idea that GLR glia have merged astrocyte and endothelial functions.

## Methods

### *Caenorhabditis elegans* strains and handling

Animals were grown on nematode growth media (NGM) plates seeded with *E. coli* (OP50) bacteria as a food source unless otherwise mentioned. Strains were maintained by standard methods (Brenner, 1974). The wild-type is strain N2, *C. elegans* variety Bristol RRID:WB-STRAIN:WBStrain00000001. A complete list of strains generated and used in this study is listed below (Table 1). A few of the strains were previously published, and/or obtained from the Caenorhabditis Genetic Center (CGC), or the TransgeneOme project.

### CRISPR/Cas9 genome editing

CRISPR/Cas9 genome editing was performed using Cas9, tracrRNAs, and crRNAs from IDT, as described in (Dokshin et al, 2018).

Generation of deletion alleles was performed by use of two crRNAs and a ssODN donor as follows:

- *unc-30(ns998[\*ns959])* [deletion of all three *let-381* motifs from the 5th intron of *unc-30*]:
  crRNAs (tctcgtgtggtataaacaat, actcggggtacagataacta) ssODN (gtcaggtaagcagaaggcaggcatcaggagttaattgggaacataattaagaatgaaaaaatatatcaaca)
- *let-381(ns1026[\*ns995])* [deletion of *let-381* autoregulatory motif]:
  crRNAs (tggttgaagagacatacatc, ttatggatggaaaacagacg) ssODN (tcatcatactttccctctatcttctcaaccagatctgtttccatccataagccaccacccattctgc). CRISPR/Cas9 generated indel: deletion from −481 to −340 (housing the tgtttata *let-381* motif) and random insertion of a 34 bp sequence cttatcttctcaatcttctcaaccagatgtgttg.
- *hlh-1(ns1015[\*syb3025])* [mutation of the 1st *let-381* motif at +726 from tgtttaca to ccgcgg and a deletion from +743 to +807 containing the 2nd *let-381* motif at +765)
  crRNAs (gtgtttacattgtgcaaact, tcttgaaaaattcgtagact) ssODN (atgggaatagtaaaggagggggggtgccgcggttgtgcaaactgggttaacccgttgtaaacataaatcgctaataggaa)

  *let-381* motif substitutions were performed by a single crRNA and a ssODN donor containing the desired mutations.
- *nep-2(ns1012[\*syb4689])* [substitution of *let-381* motif at −1133 tgtttaca to ccgcgg]
  crRNA (caattgaggaacactgggcg) ssODN (cattccgattcccacttggcactgtgccaagttgcgcccagtgttcctcaattgccgcggacagcggctccggggggc)
- *pll-1(ns1040[\*syb5792])* [substitution of *let-381* motif at −100 from tctaaata to ccgcggta]
  crRNA (acattttggcgtcgacggcg) ssODN (tttctagtagtagcaacagctcaagacattttgaagcgccgtcgacgccaaaccgcggtagaaaagaagaaaaaggaaaaaaactggaaacgg)
- *hlh-1(ns1016[\*syb3025])* [substitution of the 1st *let-381* motif +726 from tgtttaca to ccgcgg]
  crRNA (gtgtttacattgtgcaaact) ssODN (atgggaatagtaaaggagggggggtgccgcggttgtgcaataagccttttaatccatttttagtttattttccttttttcttt)
- *hlh-1(ns1027[\*syb3025])* [substitution of the 2nd *let-381* motif at +765 from agtttatt to agccatgg]
  crRNA (gtgtttacattgtgcaaact) ssODN (aggggagaagagaatttatgaaatgggtcatgggaatagtaaaggagggggggtgtttacattgtgcaaacttttgcctttaatccatttttagccatggtcctttttcttttcaattcttgaaaaattcgtagactg)
- *inx-18(ns1010[\*syb2879])* [substitution of the *let-381* motif at +364 from tctaaaca to aactgg]
  crRNA (ggtcatttctcataggaaga) ssODN (aaacttcttgacattttttggtcatttctcataggaagacttgatttccatggaaacatttttgggcggcggcgggct)
- *unc-30(ns1000[\*ns959])* [substitution of the 1st *let-381* motif]
  crRNA (tctcgtgtggtataaacaat) ssODN (cagtcaggtaagcagaaggcaggcatcaggagaaaattgtttataccacacgagaatctagaacagtgtcagttttcttcgcccccttcttg)

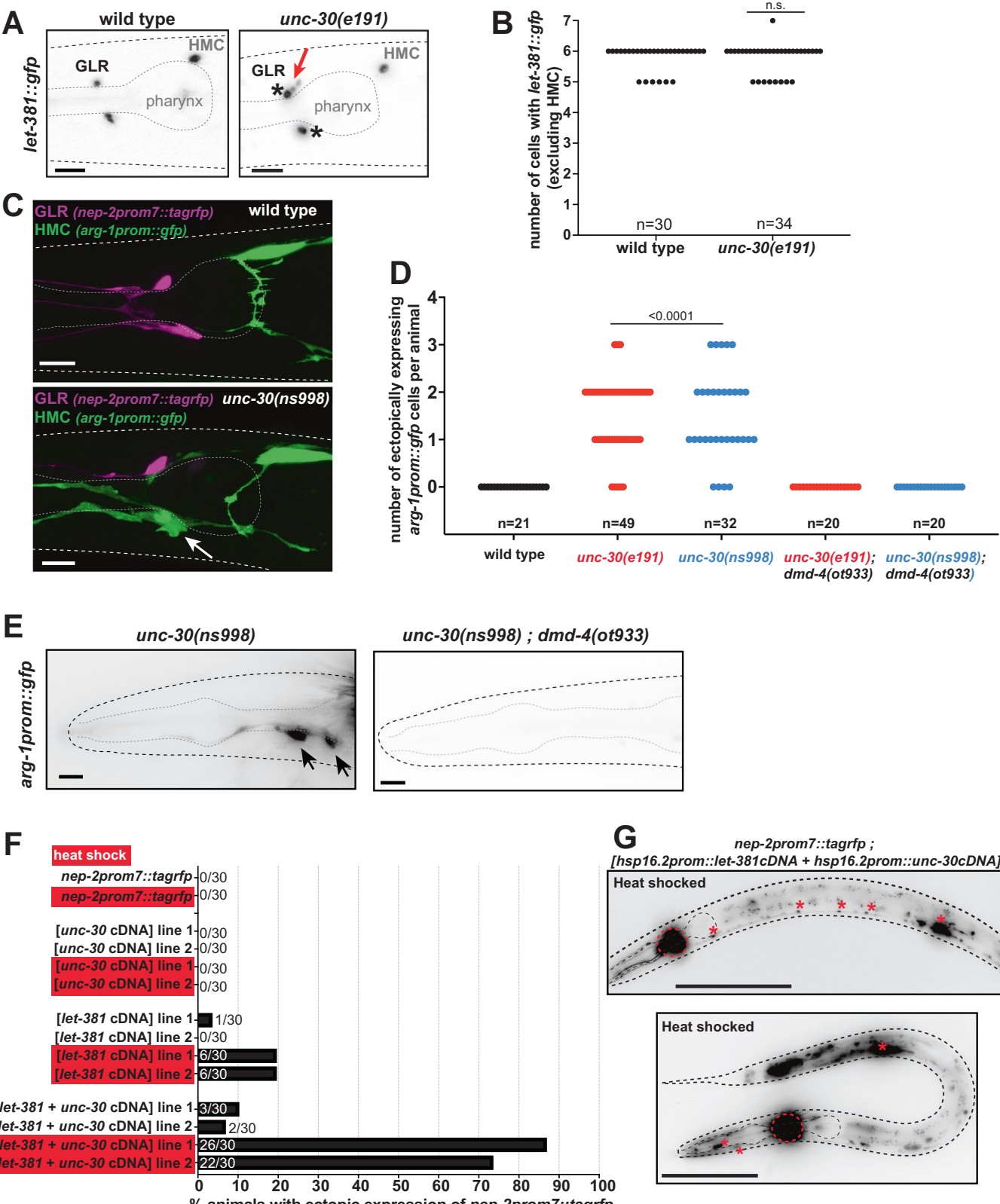

**Figure 8.** *unc-30* **represses HMC gene expression in GLR glia.**

(A) Images showing *let-381::gfp* expression in wild type (left) and *unc-30(e191)* mutants (right). Expression is observed in GLR glia anterior to the pharynx bulb and the HMC above and posterior to the pharynx bulb. In wild-type background animals, GLR glia have a small sesame-like nucleus shape. In contrast in *unc-30* mutants, some GLR glia nuclei appear larger and more round (black asterisks), reminiscent to the nucleus of the HMC cell. In addition, GLR glia are often mispositioned along the dorsoventral or left-right axis in *unc-30* mutants. Red arrow points to dorsally mispositioned cells. (B) Number of *let-381::gfp* expressing cells is unaffected in *unc-30(e191)* null mutants. (C) Expression of GLR-specific *nep-2prom7::tagrfp* (magenta) and HMC-specific *arg-1prom::gfp* (green) in wild-type and *unc-30(ns998)* mutant backgrounds. Fluorescence images of L4 animals are shown. In the GLR-specific *unc-30(ns998)* mutant background, GLR glia lose GLR-specific RFP expression and ectopically express HCM-specific GFP instead (white arrow). (D) Quantification of ectopic expression of the HMC-specific *arg-1prom::gfp* in GLR glia in *unc-30(e191)* and *unc-30(ns998)* mutants. (E) Ectopic *arg-1prom::gfp* expression in *unc-30* mutants is not observed in the *dmd-4(ot933)* mutant background. Black arrows point to ectopic *arg-1prom::gfp* expression. (F) Percentage of animals displaying ectopic expression of the GLR glia-specific reporter *nep-2prom7::rfp*, upon heat-shock-induced misexpression of *let-381* and *unc-30*. Heat-shocked animals (red boxes) are compared to age-matched non-heat-shocked controls. (G) Images showing animals with ectopic *nep-2prom7::rfp* expression after heath shock-induced misexpression of LET-381 and UNC-30. Red dashed circles outline the expression GLR glia. Red asterisks point to expression in ventral nerve cord motor neurons and stomatointestinal muscle (upper panel) and body wall and pharynx muscle (bottom panel). Data information: unpaired *t* test used for statistical analysis in (B, D). Anterior is left, dorsal is up, and scale bars are 10 µm for (A, C, E). Scale bars are 100 µm for (G). Source data are available online for this figure.

- *unc-30(ns999[\*ns959])* [substitution of the 1st + 2nd *let-381* motifs] crRNA (tctcgtgtggtataaacaat) ssODN (cagtcaggtaagcagaaggcagg-catcaggagaaaattggtaccaccacacgagaatctagaacagtgtcagttttctttcgcc ccttcttg)
- *unc-30(ns1001[\*ns959])* [substitution of the 3rd *let-381* motif]
- crRNA (cctcttatgtcataaacaattgg) ssODN (ctttcgccccttcttgccagacat-catcttgaatcttatgtcccatggattggaggcggggggtactccgcttgtctcaac)

Generation of the *let-381* and *unc-30* knock-in reporters was performed by a single crRNA and a dsDNA asymmetric hybrid-donor containing a linker::gfp::aid cassette for C-terminal tagging (see sequence below) and 120 bp flanking arms with homology to target sequence.

- *let-381(ns995[let-381::gfp::aid])* crRNA (gctattccacaagatttat)
- *unc-30(ns959[unc-30::gfp::aid])* crRNA (aagtggtccactgtactgac)
  Cassette sequence:
  **GGAGCATCGGGAGCCTCAGGAGCATCG**ATGAGTAAAG GAGAAGAACTTTTCACTGGAGTTGTCCCAATTCTTGTTG AATTAGATGGTGATGTTAATGGGCACAAATTTTCTGTCA GTGGAGAGGGTGAAGGTGATGCAACATACGGAAAACTT ACCCTTAAATTTATTTGCACTACTGGAAAACTACCTGTT CCATGGgtaagtttaaacatatatatactaactaaccctgattatttaaattttcagCCA ACACTTGTCACTACTTTCTgTTATGGTGTTCAATGCTTcT CgAGATACCCAGATCATATGAAACgGCATGACTTTTTCAA GAGTGCCATGCCCGAAGGTTATGTACAGGAAAGAACTAT ATTTTTTCAAAGATGACGGGAACTACAAGACACgtaagtttaaac agttcggtactaactaaccatacatatttaaattttcagGTGCTGAAGTCAAGTT TGAAGGTGATACCCTTGTTAATAGAATCGAGTTAAAAG GTATTGATTTTAAAGAAGATGGAAACATTCTTGGACACA AATTGGAATACAACTATAACTCACACAATGTATACATCA TGGCAGACAAACAAAAGAATGGAATCAAAGTTgtaagtttaa acatgatttactaactaactaatctgatttaaattttcagAACTTCAAAATTAGAC ACAACATTGAAGATGGAAGCGTTCAACTAGCAGACCATT ATCAACAAATACTCCAATTGGCGATGGCCCTGTCCTTT TACCAGACAACCATTACCTGTCCACACAATCTGCCCTTT CGAAAGATCCCAACGAAAAGAGAGACCACATGGTCCTT CTTGAGTTTGTAACAGCTGCTGGGATTACACATGGCAT GGATGAACTATACAAA***CCTAAAGATCCAGCCAAACCT CCGGCCAAGGCACAAGTTGTGGGATGGCCACCGGTGA GATCATACCGGAAGAACGTGATGGTTTCCTGCCAAAAT CAAGCGGTGGCCCGGAGGCGGCGGCGTTCGTGAAG***

**bold** = linker

non-bold, non-italicized = gfp with introns

***bold + italicized*** = auxin-inducible degron

Generation of the *fkh-9* knock-in reporter was performed as previously described in (Goudeau et al, 2021). Cas9, tracrRNA, crRNA and ssODN carrying the 2x(sfGFP₁₁) tag were injected, as described in (Dokshin et al, 2018), into strain CF4587 which ubiquitously expresses sfGFP(1-10). crRNA (ccagtcttttcttcttcaat), ssODN (ctttctccaatcaaggccgagagccagtcttttcttcttcaaggaggaggatccggt-gattccggcggcgttgacgtactcgtggaggaccatgtggtcacgtcctcctcccgtgacca-catggtcctccacgagtacgtcaacgccgccggaatcacctagacaccgattctgaactgatt-gaattagtcgcagttacg)

The following knock-in reporter alleles were generated by SUNY Biotech:
- *hlh-1(syb3025[hlh-1::gfp::aid])* C-terminal tagging with *linker::gfp::aid*.
- *inx-18(syb2879[inx-18::aid::emgfp])* tagged with *aid::emgfp* at position +6629 bp from *inx-18* start codon.
- *nep-2(syb4689[gfp::h2b::sl2::nep-2])* N-terminal tagging with *gfp::h2b::sl2*.
- *pll-1(syb5792[pll-1::sl2::gfp::h2b])*, *gbb-1(syb5704[gbb-1::sl2::gfp:: h2b])*, *gbb-2(syb5759[gbb-2::sl2::gfp::h2b])* C-terminal tagging with *sl2::gfp::h2b*.

## Generation of transgenic reporters and rescuing constructs and neuron identification

All reporter gene fusions for the *cis*-regulatory analysis and the minimal promoters containing *let-381* motifs were generated by a PCR fusion approach (Hobert, 2002). Genomic promoter fragments were fused to *gfp* or *tagrfp* followed by the *unc-54* 3' UTR. Promoters were initially amplified with primers A (forward) and B (reverse) and gfp or tagrfp was amplified by primers C (forward) and D (reverse). For the fusion step, amplification was done using primers A* and D* as previously described (Hobert, 2002). PCR fusion DNA fragments were injected as simple extrachromosomal arrays in a *pha-1(e2123)* mutant background strain in the following concentrations: promoter fusion (50 ng/µL), *pha-1* rescuing plasmid (pBX) (50 ng/µL). Promoter size and distance from the start codon for each promoter is shown in Fig. 1C for *nep-2*, Appendix Fig. S1A–E for *lgc-55, egl-6, gly-18, hlh-1, inx-18*, respectively, and

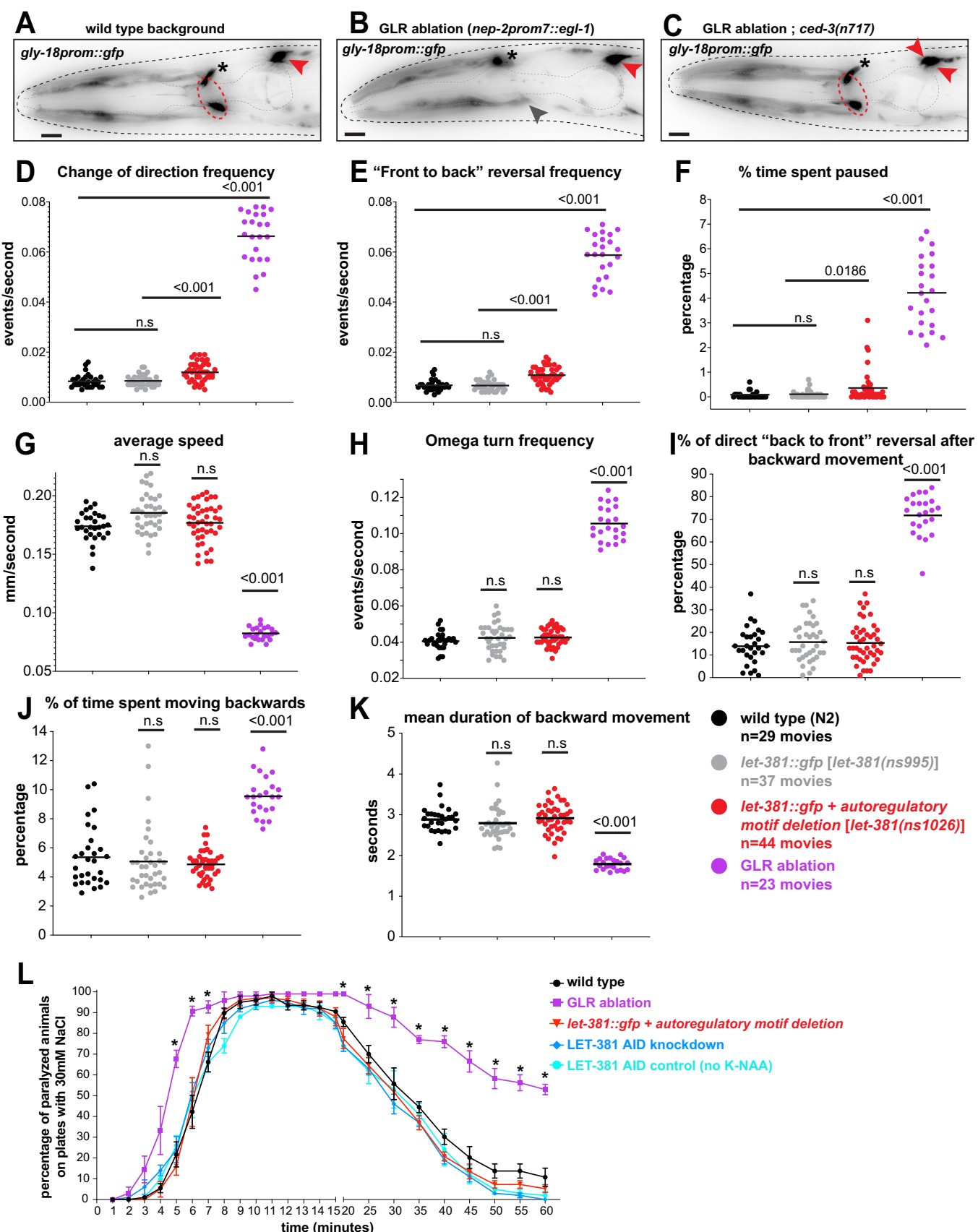

**Figure 9.   GLR glia-defective animals display locomotion abnormalities and hypersensitivity at high-salt concentrations.**

(A–C) Images of L4 animals showing expression of *gly-18prom::gfp* reporter in (A) wild-type, (B) GLR glia ablation and (C) GLR glia ablation; *ced-3(n717)* backgrounds. GLR glia are outlined by red dashed circles. Black asterisks denote expression in a neuronal cell just above the dorsal GLR glia and red arrowheads point to HMC cells. In the GLR glia-ablation background, the nerve ring is anteriorly displaced as noted by a head muscle arm (gray arrowhead) penetrating the nerve ring at the anterior pharynx bulb. The HMC sister cell that normally dies by programmed apoptotic cell death in a wild-type animal, survives and also expresses *gly-18prom::gfp* in the *ced-3(n7171)* mutant background in (C). (D–K) Locomotion parameters of foraging wild-type (black), control (gray), GLR glia-defective (red), and GLR glia-ablated (purple) animals were analyzed using automated tracking (Katz et al, 2018; Katz et al, 2019). (D) change of direction frequency, (E) "front to back" reversal frequency, (F) % time spent paused, (G) average speed, (H) omega turn frequency, (I) % direct "back to front" reversal after backward movement, (J) % time spent moving backwards, (K) mean duration of backward movement. Black lines in dot plots indicate averages. (L) GLR glia-ablated animals paralyze at a significantly higher rate than wild-type animals and animals in which LET-381 is downregulated [*let-381(ns1026)* and LET-381 AID knockdown] when exposed to 300 mM NaCl. GLR glia-ablated animals also recover motility at a significantly lower rate. Data information: unpaired *t* test was used for statistical analysis in (D–L), *<0.05 in (L). For (L), 4 replicates were performed per genotype, $n = 20$–25 for replicate, error bars indicate standard error of the mean. Anterior is left, dorsal is up, and scale bars are 10 μm for all animal images. Source data are available online for this figure.

Fig. EV3H for *unc-30*. Standard 20 bp oligos were used to amplify these promoters from wild-type (N2) genomic DNA. Oligos are listed in Appendix Table S1.

The minimal GLR glia-specific *nep-2prom7* (amplification primers: cccattccgattcccacttg, cttaaaatatgatgaggtcg) was subcloned by Gibson cloning into the *TIR1* coding plasmid pLZ31 (Zhang et al, 2015). *nep-2p7::TIR1::unc-54 3'UTR* was amplified by PCR and injected at 50 ng/μL with *unc-122::mCherry* (50 ng/μL) as co-injection marker. Similarly, *nep-2prom7* was subcloned by Gibson cloning into the *egl-1* expressing plasmid pTB28 (Bacaj et al, 2008). *nep-2prom7::egl-1::unc-54 3'UTR* was amplified by PCR and injected at 50 ng/μL with *myo-3prom::mCherry* (50 ng/μL) as co-injection marker.

Rescuing *let-381* fosmids were injected as simple extrachromosomal arrays at 50 ng/μL with a *myo-3prom::mCherry* co-injection marker (25 ng/μL) and pBluescript (25 ng/μL) as filler DNA to reach a minimum DNA injection concertation of 100 ng/μL.

Rescuing *unc-30* fosmids were linearized using NotI (NEB R3189S) and injected as complex extrachromosomal arrays in the following concentrations: linearized fosmid (30 ng/μL), linearized *unc-122prom::gfp* (5 ng/μL) as co-injection marker, sonicated *E. coli* (strain OP50) genomic DNA (100 ng/μL).

Rescuing "*unc-30* whole locus", including a 2.8kB promoter, *unc-30* gene with introns and 3'UTR, was PCR amplified from genomic DNA (primers ctgttgctcgaaaacttccg, gattaggtagaaggtagaga) and injected as simple extrachromosomal arrays at 50 ng/μL with *unc-122prom::gfp* (40 ng/μL) as co-injection marker.

Integrated transgenic constructs (*nsIs700*, *nsIs746*, *nsIs758*, *nsIs831*, *nsIs835*, *nsIs879*, *nsIs854*) were generated by exposure to 33.4 μg/mL trioxsalen (Sigma T6137) and UV irradiation using a Stratagene Stratalinker UV 2400 Crosslinker ($360 \, \mu J/cm^2 \times 100$) as previously described (Kage-Nakadai et al, 2014).

Fosmid reporters were generated as previously described (Tursun et al, 2009). More specifically, *gly-18* and *gpx-8* fosmids (WRM0641dF02 and WRM065dE01, respectively) were tagged at the C-terminus of the target genes with SL2::YFP::H2B amplified from pBALU23 (Stefanakis et al, 2015). Fosmid reporters were linearized using NotI and injected in the *pha-1(e2123)* mutant background strain as complex extrachromosomal arrays in the following concentrations: linearized fosmid (30 ng/μL), linearized pBX (pha-1 rescuing plasmid) (2.5 ng/μL), sonicated *E.coli* (strain OP50) genomic DNA 100 ng/μL.

Colocalization with the NeuroPAL transgene *otIs669* as previously described (Yemini et al, 2021), was used to identify the neurons expressing *unc-30::gfp*.

## Temporally controlled, GLR glia-specific LET-381 protein degradation

Exposure of *C. elegans* to a synthetic auxin analog K-NAA results in ubiquitination and subsequent proteasomal degradation of auxin-inducible degron (AID)-tagged proteins (Martinez et al, 2020). This occurs only in the presence of transgenically provided TIR1, the substrate recognition component of the E3 ubiquitin ligase complex. Strains carrying *let-381::gfp::aid* [*let-381(ns995)*] were crossed into a strain expressing *TIR1* specifically in GLR glia (*nsIs879* [*nep-2prom7::TIR1*]). K-NAA, 1-napthaleneacetic Acid Potassium Salt (PhytoTech Labs #N610), was dissolved in sterile $H_2O$ to prepare a 200 mM stock solution. OP50-seeded NGM plates were coated with K-NAA for a final concentration of 4 mM. Synchronized populations of L1 or late-L4/young-adult animals were transferred on K-NAA plates and grown at 20 °C for the duration of the experiment. Age-matched animals placed on OP50-seeded NGM plates coated with sterile $H_2O$, were used as controls.

## Cell-specific RNAi

A *let-381* fragment, spanning from +421 to +1578 from the *let-381* start codon (Fig. 2A), was amplified by PCR and fused in the sense and antisense orientation under the GLR glia-specific promoter *nep-2prom7* (method described in (Esposito et al, 2007)). These sense and antisense *let-381* fragments were injected at 40 ng/μL each, together with 25 ng/μL of the co-injection marker *myo-3prom::mCherry* driving expression in body wall muscle. Animals injected with unfused *nep-2prom7* (40 ng/μL) together with the co-injection marker *myo-3prom::mCherry* (25 ng/μL) were used as controls. Three independent extrachromosomal arrays were scored for each genotype. GLR glia are refractory to RNAi; thus, RNAi-sensitized background strains carrying *eri-1(mg366)* (Kennedy et al, 2004) were used for these experiments.

## Mosaic analysis

*unc-30(e191)* ; *nsIs746* [*nep-2prom7::gfp*] animals were injected with extrachromosomal arrays consisting of NotI digested *unc-30* fosmid WRM0617aE02 (30 ng/μL), *rab-3prom1::2xnls::tagrfp* (panneuronal::rfp) plasmid (35 ng/μL), *unc-122::rfp (coelomocyte::rfp)* plasmid (35 ng/μL) and sonicated *E. coli* (strain OP50) gDNA (80 ng/μL). Once extrachromosomal lines were established, mosaic animals

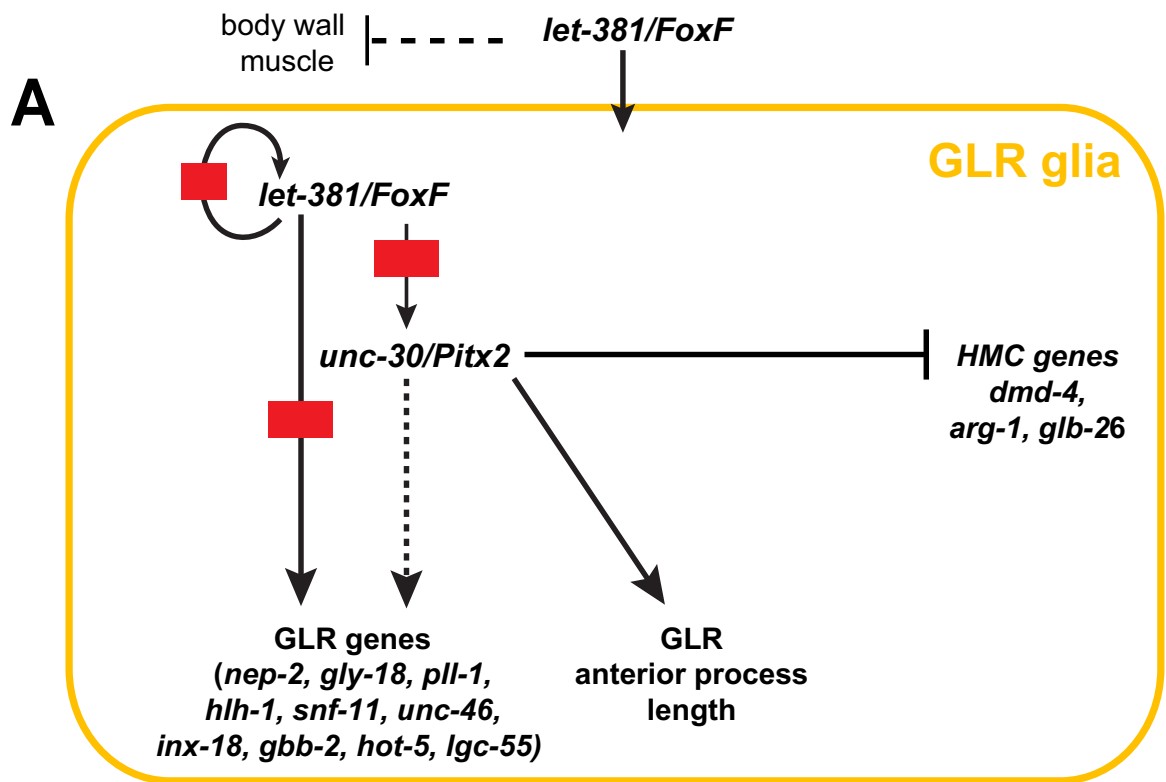

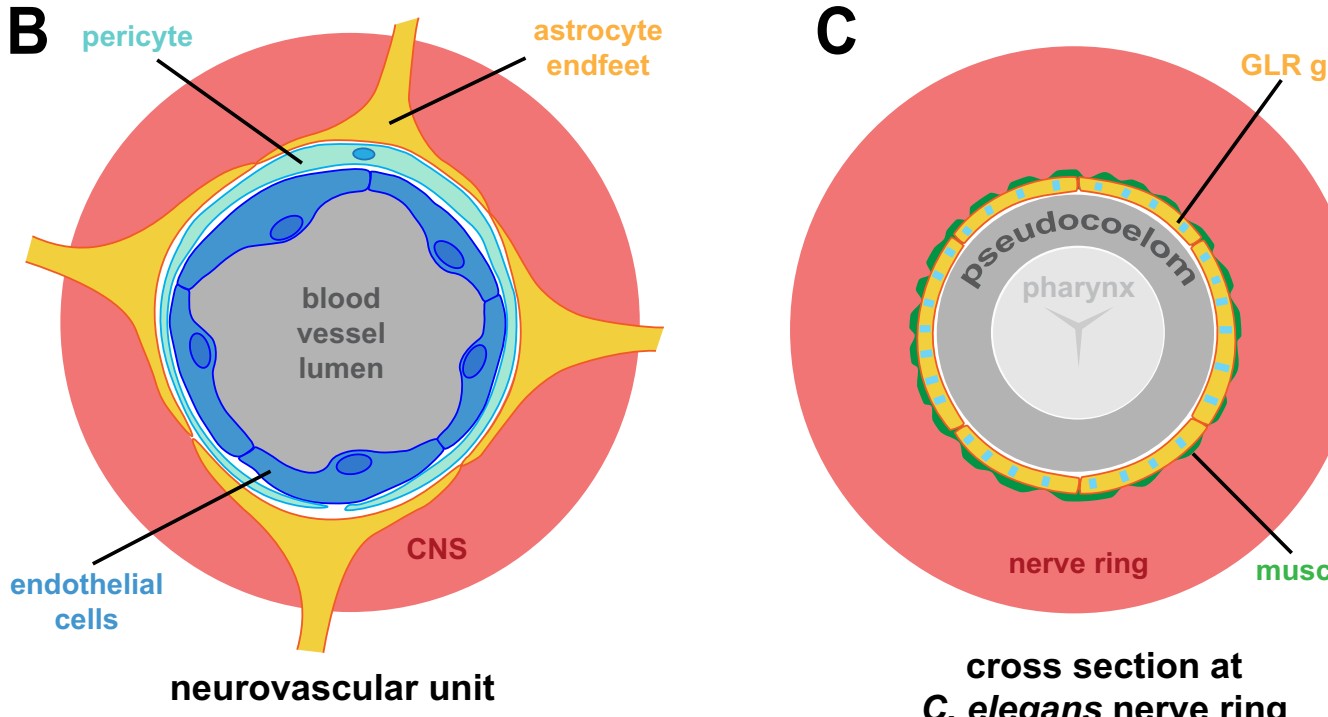

**neurovascular unit**

**cross section at
*C. elegans* nerve ring**

carrying the rescuing array in the MS lineage or the AB lineage were picked based on RFP expression under a dissecting fluorescence microscope. Rescue of the *unc-30(e191)* mutant phenotype on GLR gene expression (*nsIs746*) was then assessed by imaging of mosaic animals on a confocal microscope.

**Heat-shock-induced misexpression**

One step RT-PCR (Invitrogen #12594025) was used to isolate *let-381* and *unc-30* cDNAs (primers for *let-381*: atggaatgctcaacag, ctagcaatccgataaatc, primers for *unc-30*: atggatgacaatacggccac, ctaaagtggtccactgtact),

**Figure 10. Regulatory network for the specification and identity maintenance of the *C. elegans* GLR glia.**

(A) Schematic of the regulatory network identified in this study controlling fate specification and differentiation of GLR glia. (B) At the neurovascular unit, endothelial cells (dark blue), pericytes (light blue), and astrocytic endfeet (yellow) form the Blood–Brain Barrier isolating the central nervous system (CNS—red) from blood circulation (gray). (C) Similarly, the GLR glia sheet-like processes (yellow with blue stripes to show a mixed astrocytic-endothelial/mural fate) isolate the *C. elegans* nerve ring (red) from the pseudocoelom (gray). A thin layer of head muscle arms (green) penetrates the *C. elegans* nerve ring and therefore GLR flat processes are in close proximity to head neuromuscular junctions (White et al, 1986). The pseudocoelom is shown larger than its actual volume.

which were subsequently cloned under the heat-shock inducible promoter *hsp-16.2* by Gibson cloning. These constructs were injected at 50 ng/μL to generate transgenic lines carrying *hsp-16.2::let-381* cDNA (*nsEx6533, nsEx6534*), or *hsp-16.2::unc-30* cDNA (*nsEx6535, nsEx6536*) or both (*nsEx6530, nsEx6531*) and crossed into a strain expressing RFP specifically in GLR glia (*nsIs700 [nep-2prom7::-tagrfp]*). Twenty-five 1-day adult animals (P0) from each genotype were heat-shocked by incubating parafilm plates in a 32 °C water bath for 2.5 h. Heat-shocked P0s were kept at 25 °C and allowed to lay progeny (F1s) for 15 h before being removed from the plate. F1 progeny were grown at 25 °C for another 24 h and then scored for ectopic expression of the GLR glia-specific reporter. Age-matched, non-heat-shocked animals were used as controls.

## High-salt motility assays

One day adult animals (24 h after L4 stage, blinded for each genotype) were placed on plates without food, containing 300 mM NaCl and scored for their motility over a period of 60 min: every 1 min for the first 15 min and every five minutes after that. Animals were score for loss of motility (paralysis) and recovery from paralysis. Twenty to twenty-five animals were tested for each genotype in four replicate experiments.

## Analysis of animal locomotion

Locomotion of wild-type, GLR glia-ablated (*nsIs854*), GLR glia-defective *let-381(ns1026)*, and control *let-381(ns995)* animals were recorded and analyzed as described in detail in (Katz et al, 2018; Katz et al, 2019). Briefly, 20–40 1-day adult animals were picked to an unseeded plate, washed three times with M9 buffer, and transferred to a 6 cm plate containing 4 ml of NGM-agar without food, with a high-osmolarity barrier (4 M fructose) at the periphery of the plate to prevent wandering of animals off the plate. Animals were allowed to habituate for 20 min, then the plate was placed under a camera and locomotion was recorded for 30 min at two frames per second and thereafter analyzed as previously described (Katz et al, 2018; Katz et al, 2019). Several plates were recorded and analyzed for each genotype (29 for wild-type, 37 for control, and 44 for GLR glia-defective animals) over the span of three days. For motor behavior assays with auxin-dependent LET-381::AID down-regulation, synchronized *let-381(ns995)*; *nsIs879* L4 larvae were grown on OP50-seeded, 4 mM K-NAA plates for 24 h prior to motor behavior recordings. Locomotion assay plates also contained 4 mM of K-NAA. Age-matched animals of the same genotype, never exposed to K-NAA were used as control.

## FACS-based GLR glia glia cell isolation

For isolation of the GLR glia, synchronized L1 larvae expressing nuclear-localized YFP in the GLR glia cells (OS11715) were grown

on thirty 10-cm plates seeded with HB101 *E. coli* bacteria. After 48 h at 20 °C, L4 larvae animals were washed off plates and subsequently washed ten times with M9 to remove excess bacteria. Each wash consisted of a brief (10 s, 1300 rpm) centrifugation, such that most animals were pelleted, but bacteria stayed in suspension. Animals were then dissociated using SDS-DTT (0.25% SDS; 200 mM DTT; 20 mM HEPES, pH 8.0; 3% sucrose) and Pronase E (15 mg/ml) as previously described (Katz et al, 2019). We used 2:1 ratio of SDS-DTT to a volume of packed animals pellet, followed by 4 min incubation on ice. After five washes with ice-cold egg buffer (1.18 M NaCl; 480 mM KCl; 20 mM CaCl$_2$; 20 mM MgCl$_2$; 250 mM HEPES, pH 7.3), 4:1 ratio of room temperature Pronase E was added to the packed animals pellet and animals were then incubated rotating at 20 °C for 5 min, followed by 12 min of gentle homogenization (2 mL Dounce homogenizer, pestle clearance 0.0005–0.0025 inches, DWK Life Sciences). After three washes with ice-cold egg buffer to remove Pronase E, cells were filtered through a 5 μM filter to remove undigested animal fragments and immediately sorted by FACS.

GLR glia cell sorting was done using a BD FACS Aria sorter equipped with a 488-nm laser (Rockefeller University Flow Cytometry Resource Center), with egg buffer as the sheath buffer to preserve cell viability. Dead cell exclusion was carried out using DAPI, while DRAQ5 was used to distinguish nucleated cells from non-nucleated cell fragments. Gates for size and granularity were adjusted to exclude cell aggregates and debris. Gates for fluorescence were established using wild-type (N2) nonfluorescent animals. 500,000–800,000 YFP-positive events were sorted per replicate (4 replicates total), representing 0.3–0.7% of total events (after scatter exclusion), which is roughly the expected labeled-cell frequency in the animal (~0.6%). YFP-negative events from the same gates of size and granularity, representing all other cell types, were also sorted for comparison. Cells were sorted directly into TRIzol LS (Thermo Fisher Scientific) at a ratio 3:1 (TRIzol to cell volume).

## RNAi extraction and sequencing

RNA was extracted from the sorted cells following the TRIzol LS protocol guidelines, until the isopropanol precipitation step, then RNA was re-suspended in extraction buffer of an RNA isolation kit (PicoPure, Arcturus) and isolation continued according to the manufacturer's guideline. This two-step purification protocol helps obtain RNA of high quality when starting with samples of large volumes and resulted in a yield of around 10–80 ng per replica with RNA integrity number (RIN) ≥8, as measured by a Bioanalyzer (Agilent). All subsequent steps were performed by the Rockefeller University Genomics Resource Center. Briefly, mRNA amplification and cDNA preparation were performed using the SMARTer mRNA amplification kit (Takara #634940). Labeled samples were

**Table 1.** List of strains used and generated in this study.

| Strain name | Genotype | Comment | Reference |
|---|---|---|---|
| N2 | wild type | | CGC |
| VC706 | let-381(gk302) I/hT2 (I;III) | | (C. elegans Deletion Mutant Consortium 2012) |
| KR429 | dpy-5(e61) let-381(h107) unc-13(e450) I; sDp2 (I;f) | | (Howell et al, 1987) |
| CB845 | unc-30(e191) IV | | (Brenner, 1974) |
| VC295 | unc-30(ok613) IV | | (C. elegans Deletion Mutant Consortium 2012) |
| GR1373 | eri-1(mg366) IV | | (Kennedy et al, 2004) |
| CX3198 | sax-3(ky123) X | | (Zallen et al, 1999) |
| VC636 | cwn-2(ok895) IV | | (C. elegans Deletion Mutant Consortium 2012) |
| NG2615 | cam-1(gm122) II | | (Forrester et al, 1999) |
| DE60 | dnIs13 [gly-18prom::gfp + unc-119(+)] I I; unc-119(e2498) III | | (Warren et al, 2001) |
| MT20492 | lin-15B&lin-15A(n765) X I; nIs471 [lgc-55prom::gfp + lin-15(+)] | | (Ringstad et al, 2009) |
| OH13025 | otIs567 [unc-46(fosmid)::SL2::H2B::mCHOPTI) + pha-1(+)] | | (Gendrel et al, 2016) |
| OH13027 | otIs569 [snf-11(fosmid)::SL2::H2B::mChopti + pha-1(+)] | | (Gendrel et al, 2016) |
| PD4443 | ccIs4443 [arg-1prom::gfp + dpy-20(+)] IV | | (Kostas and Fire, 2002) |
| SD1633 | ccIs4251 [(pSAK2) myo-3p::GFP::LacZ::NLS + (pSAK4) myo-3p::mitochondrial GFP + dpy-20(+)] I; stIs10539 [dmd-4p::HIS-24::mCherry + unc-119(+)] | | (Liu et al, 2009) |
| OH16770 | ccIs4443 [arg-1prom::gfp + dpy-20(+)] IV; him-5 (e1490) V ; dmd-4(ot933) X | | (Bayer et al, 2020) |
| OP185 | unc-119 (ed3) III; wgIs185 [fkh-2-1::TY1::EGFP::3xFLAG(92C12) + unc-119(+)] | | (Sarov et al, 2006) |
| BC10849 | dpy-5(e907) I; sIs10707 [sre-6prom::GFP + dpy-5(+)] | | (McKay et al, 2003) |
| OP447 | unc-119(tm4063) III; wgIs447 [tag-68::TY1::EGFP::3xFLAG + unc-119(+)] | | (Sarov et al, 2006) |
| OH9545 | otIs287 [rab-3prom1::2xnsl::yfp, rol-6(su1006)] IV | | (Stefanakis et al, 2015) |
| OH441 | otIs45 [unc-119prom::gfp] | | (Altun-Gultekin et al, 2001) |
| CF4587 | muIs253 [eft-3prom::sfGFP1-10::unc-54 3'UTR + Cbr-unc-199(+)] II ; unc-119(ed3) III | | (Goudeau et al, 2021) |
| JN332 | peEx3332 [nep-2prom(S)::Venus, rol-6(su1006)] | | (Yamada et al, 2010) |
| OH15262 | otIs669 (NeuroPAL) | | (Yemini et al, 2021) |
| OS11153 | nsEx5558 [nep-2prom1::GFP, pha-1(+)] line 1; pha-1 (e2123) III | | This study |
| OS11157 | nsEx5562 [nep-2prom2::GFP, pha-1(+)] line 1; pha-1 (e2123) III | | This study |
| OS11158 | nsEx5563 [nep-2prom3::GFP, pha-1(+)] line 1; pha-1 (e2123) III | | This study |
| OS11162 | nsEx5567 [nep-2prom4::GFP, pha-1(+)] line 1; pha-1 (e2123) III | | This study |
| OS11163 | nsEx5568 [nep-2prom5::GFP, pha-1(+)] line 1; pha-1 (e2123) III | | This study |
| OS11165 | nsEx5570 [nep-2prom6::GFP, pha-1(+)] line 1; pha-1 (e2123) III | | This study |
| OS11166 | nsEx5571 [nep-2prom7::GFP, pha-1(+)] line 1; pha-1 (e2123) III ; otIs356 [rab-3prom1::NLS-TagRFP] V | | This study |
| OS11176 | nsEx5581 [lgc-55prom1::GFP, pha-1(+)] line 1; pha-1 (e2123) III | | This study |
| OS11177 | nsEx5582 [lgc-55prom2::GFP, pha-1(+)] line 1; pha-1 (e2123) III | | This study |
| OS11178 | nsEx5583 [lgc-55prom3::GFP, pha-1(+)] line 1; pha-1 (e2123) III | | This study |
| OS11180 | nsEx5585 [lgc-255prom4::GFP, pha-1(+)] line 1; pha-1 (e2123) III | | This study |
| OS11181 | nsEx5586 [lgc-55prom5::GFP, pha-1(+)] line 1; pha-1 (e2123) III | | This study |
| OS11182 | nsEx5587 [lgc-55prom6::GFP, pha-1(+)] line 1; pha-1 (e2123) III | | This study |
| OS11183 | nsEx5588 [lgc-55prom7::GFP, pha-1(+)] line 1; pha-1 (e2123) III | | This study |
| OS11017 | nsEx5824 [egl-6prom1::GFP, pha-1(+)] line 1; pha-1 (e2123) III | | This study |
| OS11232 | nsEx5825 [egl-6prom2::GFP, pha-1(+)] line 1; pha-1 (e2123) III | | This study |
| OS11234 | nsEx5827 [egl-6prom3::GFP, pha-1(+)] line 1; pha-1 (e2123) III | | This study |
| OS11236 | nsEx5829 [egl-6prom4::GFP, pha-1(+)] line 1; pha-1 (e2123) III | | This study |
| OS11202 | nsEx5591 [gly-18prom1::GFP, pha-1(+)] line 1; pha-1 (e2123) III | | This study |
| OS11203 | nsEx5592[gly-18prom2::GFP, pha-1(+)] line 1; pha-1 (e2123) III | | This study |
| OS11204 | nsEx5593 [gly-18prom3::GFP, pha-1(+)] line 1; pha-1 (e2123) III | | This study |
| OS11205 | nsEx5594 [gly-18prom4::GFP, pha-1(+)] line 1; pha-1 (e2123) III | | This study |
| OS11206 | nsEx5595 [gly-18prom5::GFP, pha-1(+)] line 1; pha-1 (e2123) III | | This study |
| OS11208 | nsEx5597 [gly-18prom6::GFP, pha-1(+)] line 1; pha-1 (e2123) III | | This study |
| OS11209 | nsEx5598 [gly-18prom7::GFP, pha-1(+)] line 1; pha-1 (e2123) III | | This study |
| OS11211 | nsEx5809 [gly-18prom8::GFP, pha-1(+)] line 1; pha-1 (e2123) III | | This study |
| OS11212 | nsEx5810 [gly-18prom9::GFP, pha-1(+)] line 1; pha-1 (e2123) III ; otIs356 [rab-3prom1::NLS-TagRFP] V | | This study |
| OS11168 | nsEx5573 [hlh-1prom1::GFP, pha-1(+)] line 1; pha-1 (e2123) III | | This study |

**Table 1.** (continued)

| Strain name | Genotype | Comment | Reference |
|---|---|---|---|
| OS11171 | *nsEx5576 [hlh-1prom2::GFP, pha-1(+)] line 1; pha-1 (e2123) III* | | This study |
| OS11172 | *nsEx5577 [hlh-1prom3 cloned in pSM-GFP, pha-1(+)] line 1; pha-1 (e2123) III* | | This study |
| OS11959 | *nsEx6055 [inx-18prom1::GFP, pha-1(+)] line 1; pha-1 (e2123) III* | | This study |
| OS11960 | *nsEx6056 [inx-18prom2::GFP, pha-1(+)] line 1; pha-1 (e2123)* | | This study |
| OS11962 | *nsEx6058 [inx-18prom3::GFP, pha-1(+)] line 1; pha-1 (e2123) III* | | This study |
| OS12795 | *nxEx6276 [inx-18prom4::gfp + pha-1] line 1; pha-1 (e2123) III* | | This study |
| OS12797 | *nxEx6278 [inx-18prom5::gfp + pha-1] line 1; pha-1 (e2123) III* | | This study |
| OS12798 | *nxEx6279 [inx-18prom6::gfp + pha-1] line 1; pha-1 (e2123) III* | | This study |
| OS12800 | *nxEx6281 [inx-18prom7::gfp + pha-1] line 1; pha-1 (e2123) III* | | This study |
| OS12801 | *nxEx6282 [inx-18prom8::gfp + pha-1] line 1; pha-1 (e2123) III* | | This study |
| OS12802 | *nxEx6283 [inx-18prom9::gfp + pha-1] line 1; pha-1 (e2123) III* | | This study |
| OS12819 | *nsEx6288 [inx-18prom10::gfp, pha-1+] line 1; pha-1(e2123) III* | | This study |
| OS12821 | *nsEx6290 [inx-18prom11::gfp, pha-1+] line 1; pha-1(e2123) III* | | This study |
| OS14565 | *nsEx7189 [unc-30prom1::gfp + pha-1+] line 1; pha-1(e2123) III* | | This study |
| OS14566 | *nsEx7190 [unc-30prom2::gfp + pha-1+] line 1; pha-1(e2123) III* | | This study |
| OS14567 | *nsEx7191 [unc-30prom3::gfp + pha-1+] line 1; pha-1(e2123) III* | | This study |
| OS14568 | *nsEx7192 [unc-30prom4::gfp + pha-1+] line 1; pha-1(e2123) III* | | This study |
| OS14569 | *nsEx7193 [unc-30prom5::gfp + pha-1+] line 1; pha-1(e2123) III* | | This study |
| OS14570 | *nsEx7194 [unc-30prom6::gfp + pha-1+] line 1; pha-1(e2123) III* | | This study |
| OS14571 | *nsEx7195 [unc-30prom7::gfp + pha-1+] line 1; pha-1(e2123) III* | | This study |
| OS13296 | *nsEx6463 [unc-30prom8::gfp + pha-1+] line 1; pha-1(e2123) III* | | This study |
| OS14572 | *nsEx7196 [unc-30prom9::gfp + pha-1+] line 1; pha-1(e2123) III* | | This study |
| OS14573 | *nsEx7197 [unc-30prom10::gfp + pha-1+] line 1; pha-1(e2123) III* | | This study |
| OS14094 | *nsEx6871 [F41G4.8prom1::gfp + pha-1+] line 1; pha-1 (e2123) III* | | This study |
| OS14173 | *nsEx6921 [F41G4.8prom2::gfp + pha-1+] line 1; pha-1 (e2123) III; nsIs758 [nep-2prom7::tagrf] V* | | This study |
| OS14164 | *nsEx6914 [twk-4prom1::gfp + pha-1+] line 1; pha-1 (e2123) III* | | This study |
| OS14158 | *nsEx6908 [twk-4prom2::gfp + pha-1+] line 1; pha-1 (e2123) III* | | This study |
| OS14098 | *nsEx6875 [twk-4prom3::gfp + pha-1+] line 1; pha-1 (e2123) III* | | This study |
| OS14157 | *nsEx6907 [ocr-1prom1::gfp + pha-1+] line 1; pha-1 (e2123) III* | | This study |
| OS14273 | *nsEx6966 [ocr-1prom2::gfp + pha-1+] line 1; pha-1 (e2123) III* | | This study |
| OS14097 | *nsEx6874 [ocr-1prom3::gfp + pha-1+] line 1; pha-1 (e2123) III* | | This study |
| OS14099 | *nsEx6876 [twk-9prom1::gfp +pha-1+] line 1; pha-1 (e2123) III* | | This study |
| OS14172 | *nsEx6920 [mig-6prom1::gfp + pha-1+] line 1; pha-1 (e2123) III* | | This study |
| OS14156 | *nsEx6906 [mig-6prom2::gfp + pha-1+] line 1; pha-1 (e2123) III* | | This study |
| OS14174 | *nsEx6922 [mig-6prom3::gfp + pha-1+] line 1; pha-1 (e2123) III* | | This study |
| OS14161 | *nsEx6911 [mig-6prom4::gfp + pha-1+] line 1; pha-1 (e2123) III* | | This study |
| OS14175 | *nsEx6923 [mig-6prom5::gfp + pha-1+] line 1; pha-1 (e2123) III* | | This study |
| OS14198 | *nsEx6941 [mig-17prom1::gfp + pha-1+] line 1; pha-1 (e2123) III* | | This study |
| OS14188 | *nsEx6931 [mig-17prom2::gfp + pha-1+] line 1; pha-1 (e2123) III* | | This study |
| OS14176 | *nsEx6924 [let-2prom1::gfp + pha-1+] line 1; pha-1 (e2123) III* | | This study |
| OS14159 | *nsEx6909 [let-2prom2::gfp + pha-1+] line 1; pha-1 (e2123) III* | | This study |
| OS14160 | *nsEx6910 [let-2prom3::gfp + pha-1+] line 1; pha-1 (e2123) III* | | This study |
| OS14187 | *nsEx6930 [let-2prom4::gfp + pha-1+] line 1; pha-1 (e2123) III* | | This study |
| OS14197 | *nsEx6940 [let-2prom5::gfp + pha-1+] line 1; pha-1 (e2123) III* | | This study |
| OS14276 | *nsEx6969 [acc-2prom1::gfp + pha-1+] line 1; pha-1 (e2123) III* | | This study |
| OS14268 | *nsEx6961 [oac-7prom1::gfp + pha-1+] line 1; pha-1 (e2123) III; nsIs758 [nep-2prom7::nls::yfp] V* | | This study |
| OS14307 | *nsEx6973 [kvs-5prom1::rfp + pha-1+] line 1; pha-1 (e2123) III; nsIs758 [nep-2prom7::nls::yfp] V* | | This study |
| OS14203 | *nsEx6942 [kvs-5prom2::rfp + pha-1+] line 1; pha-1 (e2123) III; nsIs758 [nep-2prom7::nls::yfp] V* | | This study |
| OS14272 | *nsEx6965 [lbp-1prom1::rfp + pha-1+] line 1; pha-1 (e2123) III; nsIs758 [nep-2prom7::nls::yfp] V* | | This study |
| OS14279 | *nsEx6972 [lbp-1prom2::rfp + pha-1+] line 1; pha-1 (e2123) III; nsIs758 [nep-2prom7::nls::yfp] V* | | This study |
| OS14308 | *nsEx6974 [adt-3prom1::rfp + pha-1+] line 1; pha-1 (e2123) III; nsIs758 [nep-2prom7::nls::yfp] V* | | This study |
| OS14343 | *nsEx7015 [R03E9.2prom1::rfp + pha-1+] line 1; pha-1 (e2123) III; nsIs758 [nep-2prom7::nls::yfp] V* | | This study |
| OS14455 | *nsEx7118 [R03E9.2prom2::rfp + pha-1+] line 1; pha-1 (e2123) III; nsIs758 [nep-2prom7::nls::yfp] V* | | This study |
| OS14345 | *nsEx7017 [R03E9.2prom3::rfp + pha-1+] line 1; pha-1 (e2123) III; nsIs758 [nep-2prom7::nls::yfp] V* | | This study |

**Table 1.** (continued)

| Strain name | Genotype | Comment | Reference |
|---|---|---|---|
| OS14346 | *nsEx7018 [R03E9.2prom4::rfp + pha-1 + ] line 1; pha-1 (e2123) III; nsIs758 [nep-2prom7::nls::yfp] V* | | This study |
| OS14398 | *nsEx7067 [F49B2.6prom1::rfp + pha-1 + ] line 1; pha-1 (e2123) III; nsIs758 [nep-2prom7::nls::yfp] V* | | This study |
| OS14399 | *nsEx7068 [F49B2.6prom2::rfp + pha-1 + ] line 1; pha-1 (e2123) III; nsIs758 [nep-2prom7::nls::yfp] V* | | This study |
| OS14395 | *nsEx7064 [nta-1prom1::rfp + pha-1 + ] line 1; pha-1 (e2123) III; nsIs758 [nep-2prom7::nls::yfp] V* | | This study |
| OS14575 | *nsEx7199 [nta-1prom2::rfp + pha-1 + ] line 1; pha-1 (e2123)III; nsIs758 [nep-2prom7::nls::yfp] V* | | This study |
| OS14400 | *nsEx7069 [T14B4.9prom1::rfp + pha-1 + ] line 1; pha-1 (e2123) III; nsIs758 [nep-2prom7::nls::yfp] V* | | This study |
| OS14458 | *nsEx7121 [T14B4.9prom2::rfp + pha-1 + ] line 1; pha-1 (e2123) III; nsIs758 [nep-2prom7::nls::yfp] V* | | This study |
| OS14483 | *nsEx7138 [T14B4.9prom3::rfp + pha-1 + ] line 1; pha-1 (e2123) III; nsIs758 [nep-2prom7::nls::yfp] V* | | This study |
| OS14459 | *nsEx7122 [tbc-12prom1::rfp + pha-1 + ] line 1; pha-1 (e2123) III; nsIs758 [nep-2prom7::nls::yfp] V* | | This study |
| OS14574 | *nsEx7198 [tbc-12prom2::rfp + pha-1 + ] line 1; pha-1 (e2123) III ; nsIs758 [nep-2prom7::nls::yfp] V* | | This study |
| OS14482 | *nsEx7137 [tbc-12prom3::rfp + pha-1 + ] line 1; pha-1 (e2123) III; nsIs758 [nep-2prom7::nls::yfp] V* | | This study |
| OS14484 | *nsEx7139 [tbc-12prom4::rfp + pha-1 + ] line 1; pha-1 (e2123) III; nsIs758 [nep-2prom7::nls::yfp] V* | | This study |
| OS14454 | *nsEx7117 [pgp-4prom1::rfp + pha-1 + ] line 1; pha-1 (e2123) III; nsIs758 [nep-2prom7::nls::yfp] V* | | This study |
| OS14457 | *nsEx7120 [pgp-4prom2::rfp + pha-1 + ] line 1; pha-1 (e2123) III; nsIs758 [nep-2prom7::nls::yfp] V* | | This study |
| OS14205 | *nsEx6944 [F19B10.3prom1::rfp + pha-1 + ] line 1; pha-1 (e2123) III; nsIs758 [nep-2prom7::nls::yfp] V* | | This study |
| OS14576 | *nsEx7200 [haf-7prom1::rfp + pha-1 + ] line 1; pha-1 (e2123); nsIs758 [nep-2prom7::nls::yfp] V* | | This study |
| OS14456 | *nsEx7119 [haf-7prom 2::rfp + pha-1 + ] line 1; pha-1 (e2123) III; nsIs758 [nep-2prom7::nls::yfp] V* | | This study |
| OS13444 | *nsEx6539 [glb-26prom1::gfp + pha-1 + ] line 1; pha-1(e2123) III* | | This study |
| OS13445 | *nsEx6540 [glb-26prom1::gfp + pha-1 + ] line 2; pha-1(e2123) III* | | This study |
| OS12140 | *nsEx6132 [WRM0641dF02 gly-18::sl2::snl::yfp::h2b, pha-1 + ] line 1; pha-1 (e2123) III* | | This study |
| OS12142 | *nsEx6134 [WRM065dE01 gpx-8::sl2::snl::yfp::h2b, pha-1 + ] line 1; pha-1 (e2123) III* | | This study |
| OS12145 | *nsEx6137[F28H7.2prom1::gfp, pha-1 + ] line 1; pha-1 (e2123) III* | | This study |
| OS12147 | *nsEx6139 [T28A11.3prom1::gfp, pha-1 + ] line 1; pha-1 (e2123) III* | | This study |
| OS12149 | *nsEx6141[ZC317.2prom1::gfp, pha-1 + ] line 1; pha-1 (e2123) III* | | This study |
| OS12151 | *nsEx6143 [F35D2.3prom1::gfp, pha-1 + ] line 1; pha-1 (e2123) III* | | This study |
| OS12789 | *nsEx6078 [F13D12.10prom1::GFP, pha-1(+)] line 1; pha-1 (e2123) III; nsIs700 V* | | This study |
| OS11967 | *nsEx6063 [mnp-1prom3::GFP, pha-1(+)] line 1; pha-1 (e2123) III* | | This study |
| OS11219 | *nsEx5817 [ncam-1prom2::GFP, pha-1(+)] line 1; pha-1 (e2123) III* | | This study |
| OS11956 | *nsEx6052 [unc-54prom3::GFP, pha-1(+)] line 1; pha-1 (e2123)* | | This study |
| OS12370 | *nsEx6202 [WRM061BG12 best-22 TY1::EGFP::3xFLAG(92C12) + pha-1(+)] line 1; pha-1 (e2123) III* | Tagged fosmid generated by (Sarov et al, 2006). Transgenic line generated in this study. | |
| OS14023 | *nsEx6821 [WRM069bF08 (let-381 fosmid) + myo-3p::mCherry] line 1; let-381(gk302) I; nsIs746 [nep-2prom7::gfp] V* | | This study |
| OS14024 | *nsEx6822 [WRM069bF08 (let-381 fosmid) + myo-3p::mCherry] line 2; let-381(gk302) I; nsIs746 [nep-2prom7::gfp] V* | | This study |
| OS12240 | *nsEx6151 [WRM0639aA08 (unc-30 fosmid) + unc-122prom::gfp)] line 1; unc-30 (e191) IV; nsIs700 [nep-2prom7::tagrfp] V* | | This study |
| OS12241 | *nsEx6152 [WRM0639aA08 (unc-30 fosmid) + unc-122prom::gfp)] line 2; unc-30 (e191) IV; nsIs700 [nep-2prom7::tagrfp] V* | | This study |
| OS12242 | *nsEx6153 [WRM0639aA08 (unc-30 fosmid) + unc-122prom::gfp)] line 3; unc-30 (e191) IV; nsIs700 [nep-2prom7::tagrfp] V;* | | This study |
| OS12246 | *nsEx6157 [WRM0617aE02 (unc-30 fosmid) + unc-122prom::rfp] line 1; unc-30 (e191) IV; nsIs746 [nep-2prom7::gfp] V* | | This study |
| OS12247 | *nsEx6158 [WRM0617aE02 (unc-30 fosmid) + unc-122prom::rfp] line 2; unc-30 (e191) IV; nsIs746 [nep-2prom7::gfp] V* | | This study |
| OS12244 | *nsEx6155 [unc-30 whole locus PCR + unc-122prom::gfp] line 1; unc-30 (e191) IV; nsIs700 [nep-2prom7::tagrfp] V* | | This study |
| OS13665 | *nsEx6668 [nep-2p7::let-381 sas RNAi + myo-3::mCherry] line 1; let-381(ns995) I; eri-1(mg366) IV* | | This study |
| OS13666 | *nsEx6669 [nep-2p7::let-381 sas RNAi + myo-3::mCherry] line 2; let-381(ns995) I; eri-1(mg366) IV* | | This study |
| OS13667 | *nsEx6670 [nep-2p7::let-381 sas RNAi + myo-3::mCherry] line 3; let-381(ns995) I; eri-1(mg366) IV* | | This study |
| OS13668 | *nsEx6671 [nep-2p7 (sas RNAi control) + myo-3::mCherry] line 1; let-381(ns995) I; eri-1(mg366) IV* | | This study |
| OS13820 | *nsEx6667 [nep-2p7 (sas RNAi control) + myo-3::mCherry] line 2 ; nep-2(syb4689) II; eri-1(mg366) IV* | | This study |
| OS13672 | *nsEx6675 [nep-2p7 (sas RNAi control) + myo-3::mCherry] line 3; ccIs4443 [arg-1prom::gfp + dpy-20(+)] eri-1(mg366) IV* | | This study |
| OS13435 | *nsEx6530 [hsp-16,2prom::let-381 cDNA, hsp-16,2prom::unc-30 cDNA, pha-1 + ] line 1; pha-1(e2123) III; nsIs700 [nep-2prom7::tagrfp] V* | | This study |
| OS13436 | *nsEx6531 [hsp-16,2prom::let-381 cDNA, hsp-16,2prom::unc-30 cDNA, pha-1 + ] line 2; pha-1(e2123) III; nsIs700 [nep-2prom7::tagrfp] V* | | This study |
| OS13438 | *nsEx6533 [hsp-16,2prom::let-381 cDNA, pha-1 + ] line 1; pha-1(e2123) III; nsIs700 [nep-2prom7::tagrfp] V* | | This study |
| OS13439 | *nsEx6534 [hsp-16,2prom::let-381 cDNA, pha-1 + ] line 2; pha-1(e2123) III; nsIs700 [nep-2prom7::tagrfp] V* | | This study |
| OS13440 | *nsEx6535 [hsp-16,2prom::unc-30 cDNA, pha-1 + ] line 1; pha-1(e2123) III; nsIs700 [nep-2prom7::tagrfp] V* | | This study |

**Table 1.** (continued)

| Strain name | Genotype | Comment | Reference |
|---|---|---|---|
| OS13441 | nsEx6536 [hsp-16,2prom::unc-30 cDNA, pha-1 + ] line 2; pha-1(e2123) III; nsIs700 [nep-2prom7::tagrfp] V | | This study |
| OS11484 | nsIs700 [nep-2prom7::tagrfp] V | | This study |
| OS11703 | nsIs746 [nep-2prom7::gfp] V | | This study |
| OS11715 | nsIs758 [nep-2prom7::nls::yfp] V | | This study |
| OS12099 | nsIs831 [pll-1prom1::tagrfp] X | | This study |
| OS12103 | nsIs835 [hot-5prom2::gfp] X | | This study |
| OS12767 | nsIs879 [nep-2p7::TIR1::mCardinal + unc-122prom::mCherry] | | This study |
| OS12156 | nsIs854 [nep-2prom7::egl-1, myo-3mCherry]; nsIs831 [pll-1prom1::tagrfp] X | | This study |
| OS12700 | unc-30(ns959[unc-30::gfp:::degron(AID)]) IV | | This study |
| OS13288 | let-381(ns995[let-381::gfp::aid]) I | | This study |
| PHX4689 | nep-2(syb4689[gfp::h2b::sl2::nep-2]) II | | This study |
| PHX3025 | hlh-1(syb3025[hlh-1::gfp::aid]) II | | This study |
| PHX2879 | inx-18(syb2879[inx-18::aid::emgfp]) IV | | This study |
| PHX5792 | pll-1(syb5792[pll-1::sl2::gfp::h2b]) III | | This study |
| PHX5704 | gbb-1(syb5704[gbb-1::sl2::gfp::h2b]) X | | This study |
| PHX5759 | gbb-2(syb5759[gbb-2::sl2::gfp::h2b]) IV | | This study |
| OS13744 | nep-2(ns1012[*syb4689]) II | ns1012 = let-381 motif mutation | This study |
| OS14214 | pll-1(ns1040[*syb5792]) III | ns1040 = let-381 motif mutation | This study |
| OS13748 | hlh-1(ns1016[*syb3025]) II | ns1016 = 1$^{st}$ let-381 motif mutation | This study |
| OS13842 | hlh-1(ns1027[*syb3025]) II | ns1027 = 2$^{nd}$ let-381 motif mutation | This study |
| OS13747 | hlh-1(ns1015[*syb3025]) II | ns1015 = 1$^{st}$ + 2$^{nd}$ motif mutation | This study |
| OS13742 | inx-18(ns1010[*syb2879]) | ns1010 = let-381 motif mutation | This study |
| OS13838 | let-381(ns1026[*ns995]) I | ns1026 = let-381 motif deletion | This study |
| OS13308 | unc-30(ns1000[*ns959]) IV | ns1000 = 1$^{st}$ let-381 motif mutation | This study |
| OS13307 | unc-30(ns999[*ns959]) IV | ns999 = 1$^{st}$ + 2$^{nd}$ let-381 motifs mutation | This study |
| OS13309 | unc-30(ns1001[*ns959]) IV | ns1001 = 3$^{rd}$ let-381 motif mutation | This study |
| OS13306 | unc-30(ns998[*ns959]) | ns998 = three let-381 motifs deletion | This study |

sequenced using an Illumina NextSeq 500 sequencer using 75 base pair single read protocols.

## RNA-seq quality assessment and differential expression analysis

Fastq files were generated with CASAVA v1.8.2 (Illumina), and examined using the FASTQC (http://www.bioinformatics. babraham.ac.uk/projects/fastqc/) application for sequence quality. Reads were aligned to customized build genome that combine *C. elegans* WS262 genome release (https://downloads.wormbase.org/ releases/WS262/species/c_elegans/PRJNA13758/) and the "*nep-2prom7::nls::::yfp::unc-54 3'UTR*" transgene using the STAR v2.3 aligner with parameters (Dobin et al, 2012) (--out- FilterMultimapN-max 10 --outFilterMultimapScoreRange 1). Mapping rate was >72% with >62 million uniquely mapped reads. The alignment results were evaluated through RNA-SeQC v1.17 to make sure all samples had a consistent alignment rate and no obvious 5' or 3' bias (DeLuca et al, 2012). Aligned reads were summarized through featureCounts (Liao et al, 2013) with gene models from Ensemble (Caenorhabditis_elegans. WBcel235.77.gtf) at gene level unstranded: specifically, the uniquely mapped reads (NH "tag" in bam file) that overlapped with an exon

(feature) by at least 1 bp on either strand were counted, and then the counts of all exons annotated to an Ensemble gene (meta-features) were summed into a single number. rRNA genes, mitochondrial genes, and genes with length <40 bp were excluded from downstream analysis.

The experiment was done with four independent replicates. DESeq2 was applied to normalize count matrix and to perform differential gene expression analysis, comparing RNA counts derived from the GLR glia cells (YFP-positive) to RNA counts that were derived from all other *C. elegans* cells, using the negative binomial distribution (Love et al, 2014). Raw and processed RNA-seq data have been deposited at Gene Expression Omnibus with accession number GEO: GSE234746.

## Microscopy

Animals were anesthetized using 100 mM NaN$_3$ (sodium azide) and mounted on 5% agarose pads on glass slides. Z-stack images (each ~7 μm thick) were acquired using either a Zeiss confocal microscope LSM990 (images in Figs. 7C and 8D; Appendix Fig. S5D,E) or a Zeiss compound microscope Axio Imager M2 (all other images) using MicroManager software (version 1.4.22) (Edelstein et al, 2010). ImageJ (Schneider et al, 2012) was used to

produce maximum projections of z-stack images (2–30 slices) presented in the Figures. Figures were prepared by using Adobe Illustrator.

## Quantification and statistical analysis

All microscopy fluorescence quantifications were done in ImageJ (Schneider et al, 2012). Mutant and control animals were imaged during the same imaging session with all acquisition parameters maintained constant between the two groups. The fluorescence intensity of gene expression in GLR glia and the HMC cell (Figs. 4E, F, 5A, B, D, and 7G) was measured in the plane with the strongest signal within the z-stack in a region drawn around the GLR glia nucleus (for nuclear reporters) or cell body (for cytoplasmic reporters). A single circular region in an adjacent area was used to measure background intensity for each animal; this value was then subtracted from the fluorescence intensity of reporter expression for each GLR glia/HMC cell. Quantification of the length of the GLR glia anterior process (Fig. 7E) was performed in maximum intensity projections. A line was drawn along the anterior process for each GLR glia cell and its length was measured and normalized to the length of the pharynx for each animal. Quantification of number of puncta for *inx-18::gfp* CRISPR knock-in reporter (Figs. 5C and 6E), quantification of number or percentage of GLR glia or HMC with reporter expression (Figs. 2C,D, 3A–G, 6A–E, 7D, 8B, D, EV2C–G, EV3A-G,J–K, and EV4C,E,F; Appendix Fig. S2B,C), quantification of animals with anteriorly displaced nerve ring (Fig. 2E) and quantification of number of head muscle cells (Fig. 2F; Appendix Figs. S2D and S3A) were performed by manual counting using ImageJ.

Prism (GraphPad) was used for statistical analysis as described in Figure legends. Unpaired, two-sided Students' *t* test was used to determine the statistical significance between the two groups. n.s. denotes not statistically significant (*P* value > 0.05).

## Data availability

Raw and processed RNA-seq data have been deposited at Gene Expression Omnibus with accession number GEO: GSE234746. This paper does not report any original code. All newly generated critical strains will be available at the Caenorhabditis Genetics Center (CGC). Any additional information required to reanalyze the data reported and requests for resources and reagents should be directed to and will be fulfilled by the corresponding author.

## Peer review information

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

## Acknowledgements

The authors thank Oliver Hobert and Yuichi Lino for strains; The Rockefeller University Flow Cytometry and Genomics Resource Centers for technical support; Menachem Katz and members of the Shaham lab for experimental advice, comments, and discussion. Some strains were provided by the CGC, which is funded by NIH Office of Research Infrastructure Programs (P40 OD010440). This work was supported in part by funds from a Leon Levy Foundation Fellowship to NS and by National Institutes of Health grant R35NS105094 to SS.

## Author contributions

**Nikolaos Stefanakis**: Conceptualization; Formal analysis; Funding acquisition; Investigation; Visualization; Methodology; Writing—original draft; Writing—review and editing. **Jessica Jiang**: Investigation. **Yupu Liang**: Software; Formal analysis. **Shai Shaham**: Conceptualization; Funding acquisition; Writing—original draft; Project administration; Writing—review and editing.

## Disclosure and competing interests statement

The authors declare no competing interests.

# Expanded View Figures

**Figure EV1.** *let-381* **motifs are required for endogenous GLR gene expression and** *let-381* **autoregulation in GLR glia.**

(**A**) *let-381* motif identified in this study. (**B**) Motif of the *let-381* ortholog *foxf* from (Peterson et al, 1997). (**C**) *let-381* motif from (Narasimhan et al, 2015). Similarities between the three motifs are apparent. (**D**) Locations (distances from start codons) of *let-381* motifs of genes whose expression in GLR is downregulated in *let-381* mutants (either GLR-specific *let-381* RNAi and/or the *let-381* autoregulatory allele). (**E**) Minimal promoter *hlh-1prom1* was one of the promoters used in MEME to identify common motifs present in GLR glia genes. The *let-381* motif identified by MEME is highlighted in dark red. A *let-381* motif with slightly altered sequence (light red) was identified manually later and is required, together with the first motif, to control *hlh-1* expression in GLR glia. (**F**) Schematics showing details on endogenous *gfp*-based tags, location of *let-381* motifs and their mutation for *nep-2*, *pll-1*, *hlh-1* and *inx-18* genes. Red bars represent *let-381* motifs. Distance from ATG is indicated above each motif. Nucleotide changes for each motif mutation is shown below the motifs.

**A**

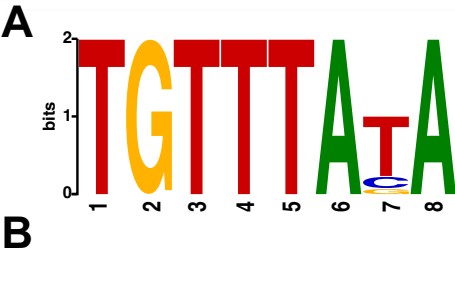

**B**

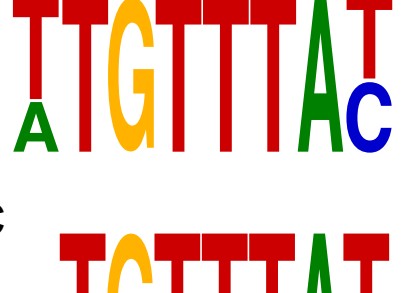

**C**

**D**

| gene name | location of *let-381* motif(s) in base pairs from start codon |
|---|---|
| *nep-2* | −1133 |
| *gly-18* | −147 |
| *hlh-1* | +726 and +765 |
| *inx-18* | +364 |
| *hot-5* | −135 and −122 |
| *pll-1* | −100 |
| *lgc-55* | −3867 |
| *unc-46* | −921 |
| *snf-11* | −840 |
| *gbb-2* | −388 |

**E**   *hlh-1p1 270bp*

ggttataatgagcaccagatgaggctatttgttctgtacaggagcctgg

cggctaggctttttgcatgctattgattaatagggaatgcggggatgg

aaaaatcgaagtagtagggaaaggggagaagagaatttatgaaatgg

gtcatgggaatagtaaagggaggggggtgtttaca ttgtgcaaacttg

ggccttttaatccatttt agtttatt tccttttctttttcaattcttgaaaaattc

gtagactgggttaacccgttg

**F**

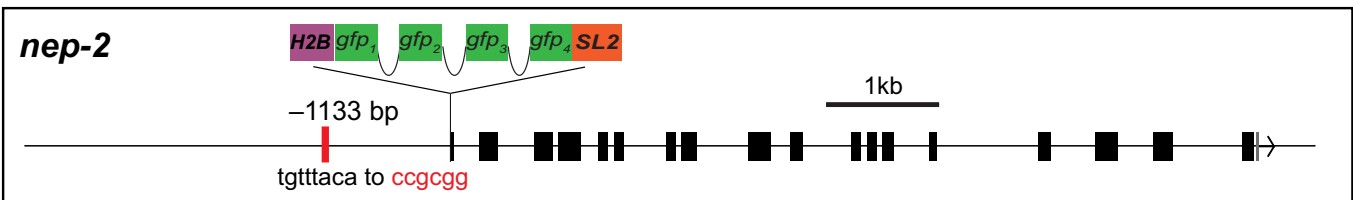

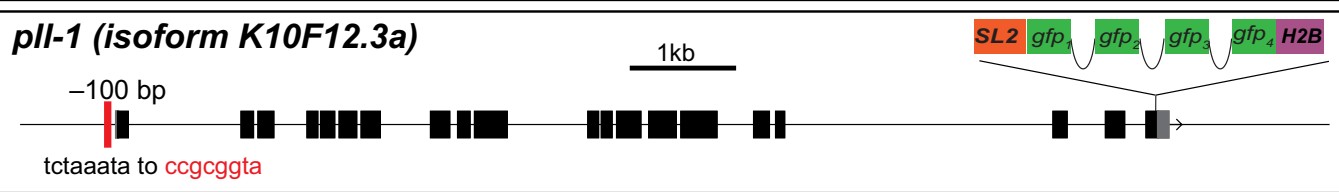

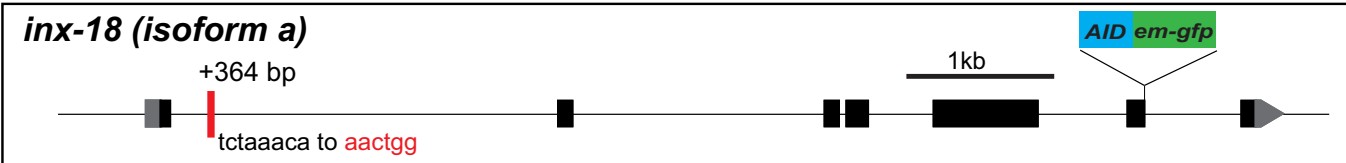

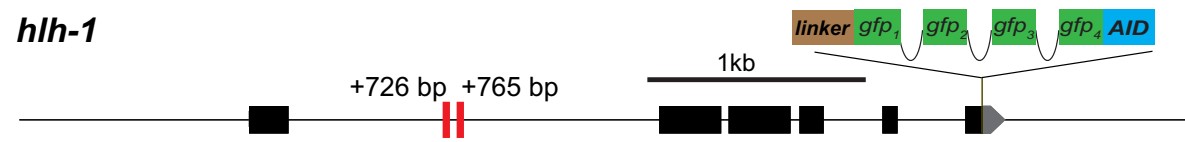

**"1st motif mutated"** is: tgtttaca to ccgcgg at +726

**"2nd motif mutated"** is: agtttatt to agccatgg at +765

**"Both motifs mutated"** is: tgtttaca to ccgcgg at +726 and a deletion from +743 to +807 removing the second motif

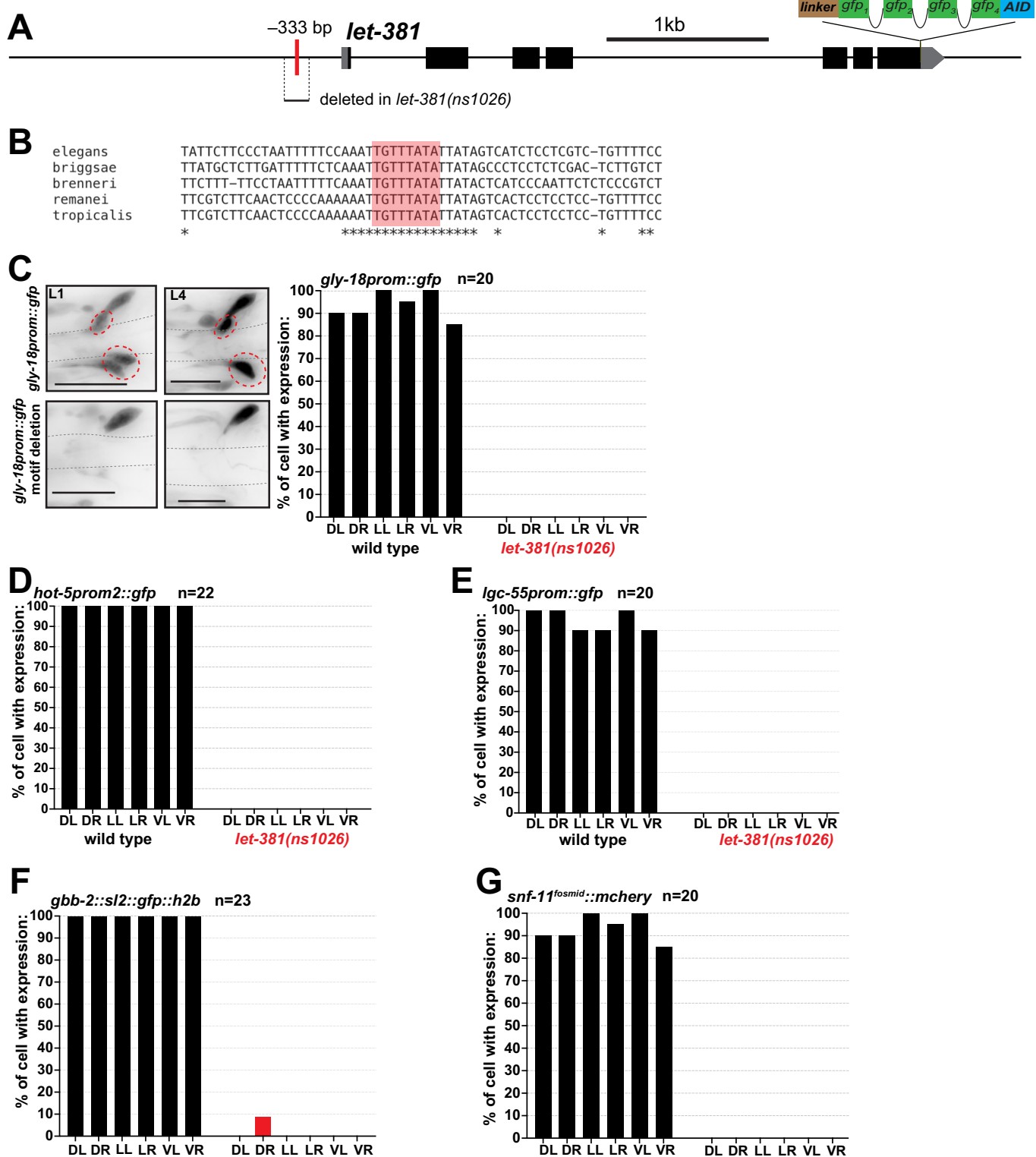

◀ **Figure EV2. GLR gene expression is lost in *let-381* autoregulatory mutant animals.**

(A) Schematic showing the location of the *let-381* motif (red bar) in the *let-381* promoter region and region deleted in the *let-381(ns1026)* mutation. (B) Conservation of the *let-381* autoregulatory motif sequence (red box) is shown among five nematode species. Asterisks indicate conserved nucleotides. (C–G) Effect of *let-381* autoregulatory motif deletion, *let-381(ns1026)*, on expression of (C) *gly-18*, (D) *hot-5*, (E) *lgc-55*, (F) *gbb-2*, and (G) *snf-11* in GLR glia. Bar graphs show quantifications of gene expression at the L4 stage. For (C) animal images showing gene expression at L1 and L4 stages in wild-type and mutant backgrounds are shown on the left. Dashed red circles outline expression in GLR glia. Data information: Anterior is left, dorsal is up and scale bars are 10 μm for all animal images.

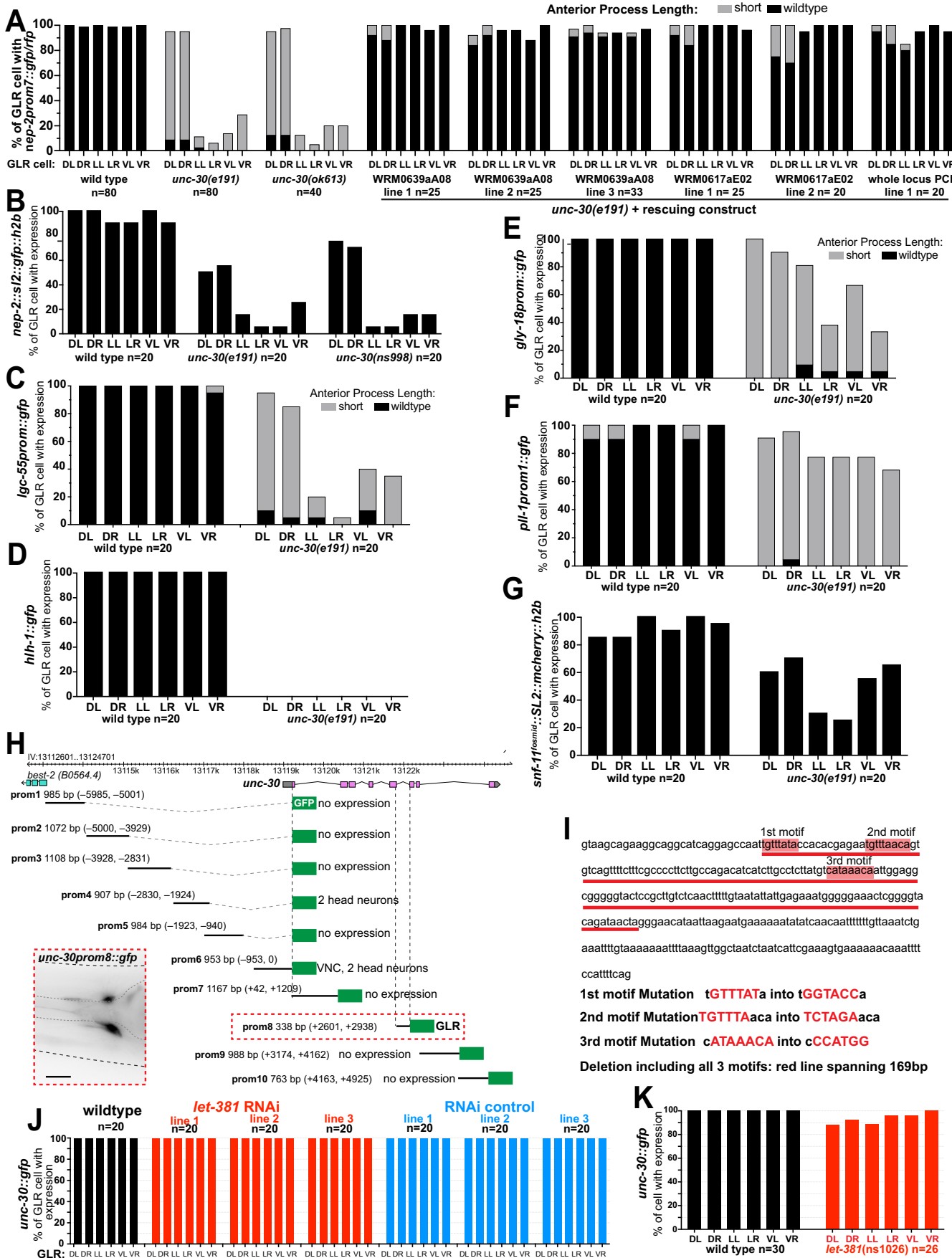

◀  **Figure EV3.  Effect of *unc-30* on GLR gene expression.**

(A) Transgenic constructs containing different fosmid clones (WRM) or PCR amplicons carrying wild-type copies of UNC-30 can rescue the effect of *unc-30(e191)* on GLR gene expression and anterior process length. (B-G) Effect of *unc-30* mutation on expression of different genes in GLR glia. Expression of (E) *gly-18*, (F) *pll-1* and (G) *snf-11* is affected at a lesser extent compared to (B) *nep-2*, (C) *lgc-55* and (D) *hlh-1*. (H) Cis-regulatory dissection analysis of *unc-30*. The fifth intron (prom8) of *unc-30* is sufficient to drive expression in GLR glia. (I) Three *let-381* motifs are found in the fifth intron of *unc-30* (red boxes). Details on *let-381* motif mutation alleles are shown below the DNA sequence. (J, K) Endogenous *unc-30::gfp* expression is not affected by postembryonic *let-381* knockdown either (J) by GLR-specific RNAi or (K) in the GLR-specific *let-381* autoregulatory motif deletion allele *let-381(ns1026)*. Data information: Anterior is left, dorsal is up and scale bars are 10 μm for all animal images.

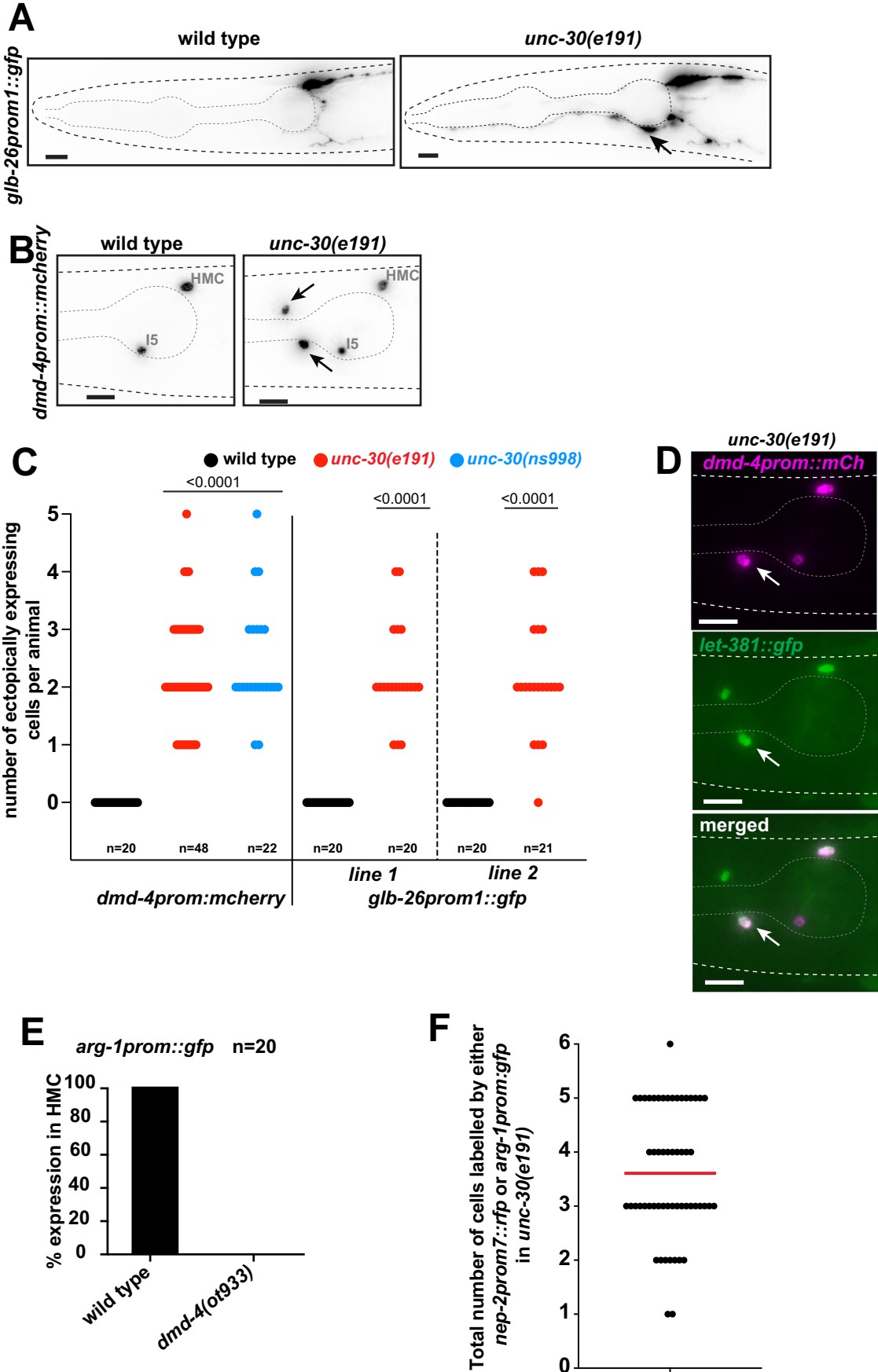

◄  **Figure EV4.  *unc-30* represses HMC gene expression in GLR glia.**

(**A**) Fluorescence images showing expression of *glb-26prom::gfp* in wild type and *unc-30(e191)* mutants. (**B**) Fluorescence images showing expression of *dmd-4prom::mCherry* in wild type and *unc-30(e191)* mutants. Arrows point to ectopic expression in *unc-30(e191)* mutants. (**C**) Quantification of ectopic expression of the two HMC reporters shown in (**A**) and (**B**) in *unc-30* mutant backgrounds. (**D**) Cells ectopically expressing (white arrow) the HMC reporter *dmd-4prom::mCherry* (magenta) always co-express *let-381::gfp* (green). (**E**) Expression of *arg-1prom::gfp* is lost in HMC in *dmd-4(ot933)* mutants. (**F**) Total number of GLR glia cells expressing either the GLR glia-specific *nep-2prom7::rfp* or the HMC-specific *arg-1prom::gfp* in *unc-30(e191)* mutants. Data information: unpaired *t* test used for statistical analysis in (**C**). Anterior is left, dorsal is up and scale bars are 10 µm for all animal images.

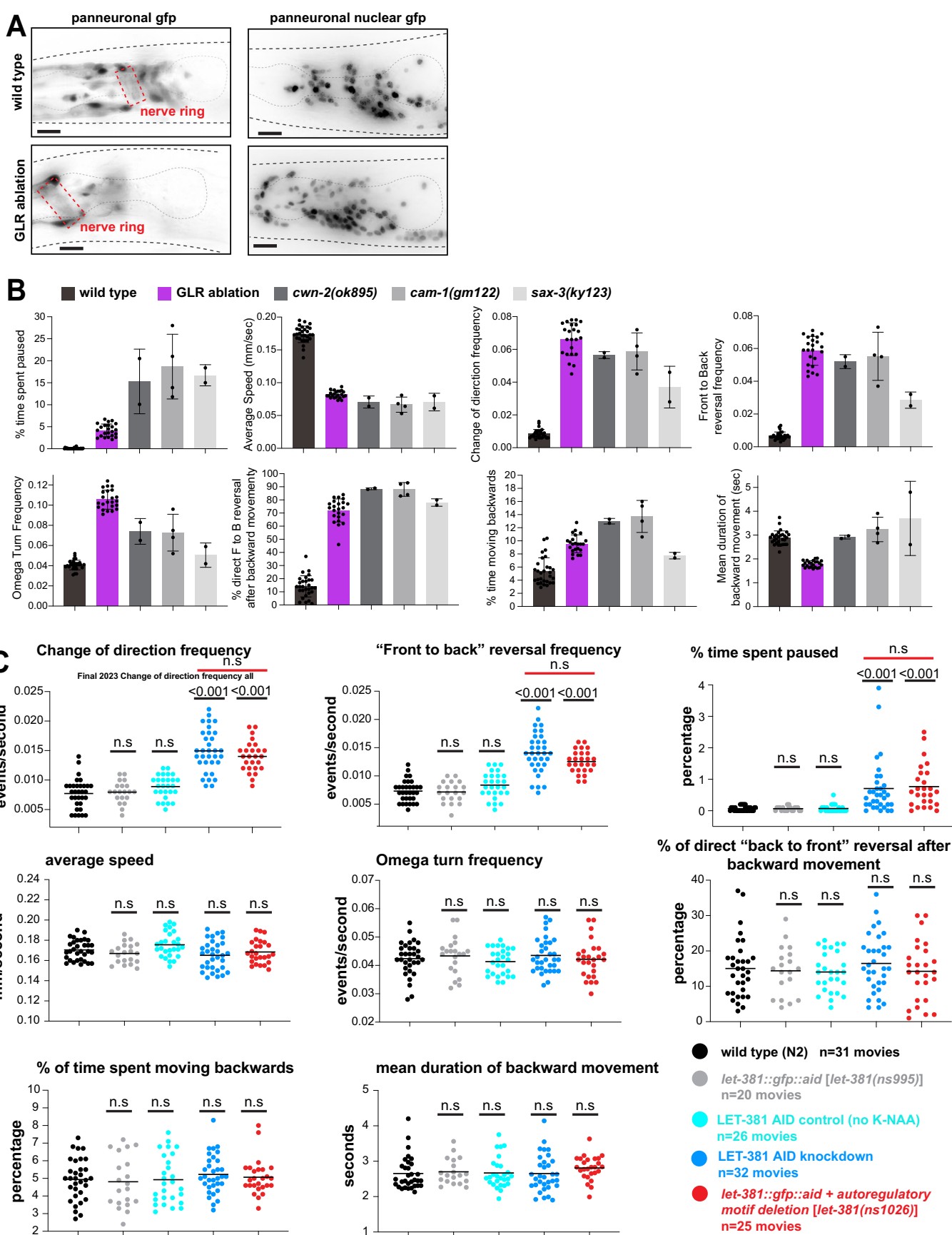

◀ **Figure EV5. Locomotion defects of GLR-ablated animals could partially be due to anteriorly displaced nerve ring.**

(A) In wild-type animals (top row), axons of the nerve ring (dashed red box) are located between the two pharyngeal bulbs. In GLR-ablated animals (bottom row) the nerve ring is anteriorly displaced, located on top of the anterior pharynx bulb. As evidenced in the images on the right, not only the axonal projections, but also neuronal cell bodies (panneuronal nuclear gfp) are anteriorly displaced. Panneuronal gfp = *unc-119prom::gfp*, panneuronal nuclear gfp = *rab-3prom1::nls::yfp*. (B) *cwn-2(ok895)*, *cam-1(gm122)*, *sax-3(ky123)* mutants with anteriorly displaced nerve rings exhibit locomotion defects to the same direction, although of different magnitude as the GLR glia-ablated animals. (C) Auxin (K-NAA) dependent LET-381::AID knockdown results in similar defects in the same locomotion parameters as the *let-381(ns1026)* autoregulatory mutation. Genotypes are: wild-type N2 (black), *let-381(ns995)* control (gray), LET-381::AID knockdown [*let-381(ns995);nsIs879 (nep-2prom7::TIR1)*] exposed to K-NAA auxin (dark blue), LET-381::AID control [*let-381(ns995) ; nsIs879 (nep-2prom7::TIR1)*] not exposed to K-NAA auxin (light blue) and *let-381(ns1026)* autoregulatory mutation (red). Data information: in (B) wild type *n* = 29 movies, GLR ablation *n* = 23 movies, *cwn-2(ok895)* *n* = 2 movies, *cam-1(gm122)* *n* = 4 movies, *sax-3(ky123)* *n* = 2 movies. Bar height indicates average (center of error bars) and error bars show standard deviation in (B). Unpaired t test used for statistical analysis in (C); controls (gray and light blue) were compared to wild type (black). LET-38::AID knockdown (dark blue) was compared to its control group (light blue) and *let-381(ns1026)* was compared to its control (gray). No statistically significant differences were observed between the LET-381::AID knockdown and *let-381(ns1026)* as indicated by the red line on the top of the three upper diagrams. Anterior is left, dorsal is up and scale bars are 10 µm for all animal images.

