## [Peer Review File · The EMBO Journal]

LET-381/FoxF and UNC-30/Pitx2 control development of mesodermal glia that regulate motor behavior

Nikolaos Stefanakis, Jessica Jiang, Yupu Liang, and Shai Shaham

Corresponding author: Shai Shaham (shaham@rockefeller.edu)

Review Timeline:

Submission Date:	7th Sep 23
Editorial Decision:	17th Oct 23
Revision Received:	22nd Dec 23
Editorial Decision:	18th Jan 24
Revision Received:	22nd Jan 24
Accepted:	26th Jan 24

Editor: Ieva Gailite

Transaction Report:

Dear Dr. Shaham,

Thank you for submitting your manuscript for consideration by the EMBO Journal. We have now received comments from three reviewers, which are included below for your information.

As you will see from the reports, all reviewers find the study of interest, while also pointing out a number of aspects that would need to be improved in the final manuscript before they can recommend acceptance. From my side, I find the comments generally reasonable. If point 1 by referee #2 cannot be addressed due to the lack of available stage-specific promoters, this will not be absolutely required for acceptance here.

Based on the interest expressed in the reports, I would like to invite you to address the issues raised by the referees in a revised manuscript. I would be happy to discuss the revision in more detail via email or phone/videoconferencing - please let me know which option you prefer.

We generally allow three months as standard revision time. As a matter of policy, competing manuscripts published during this period will not negatively impact on our assessment of the conceptual advance presented by your study. However, please contact me as soon as possible upon publication of any related work to discuss the appropriate course of action. Should you foresee a problem in meeting this deadline, please let us know in advance to discuss an extension.

When preparing your letter of response to the referees' comments, please bear in mind that this will form part of the Review Process File and will therefore be available online to the community. For more details on our Transparent Editorial Process, please visit our website: <https://www.embopress.org/page/journal/14602075/authorguide#transparentprocess>. Please also see the attached instructions for further guidelines on preparation of the revised manuscript.

Please feel free to contact me if you have any further questions regarding the revision. Thank you for the opportunity to consider your work for publication. I look forward to discussing your revision.

With best regards,

Ieva

- a point-by-point response to the referees' comments, with a detailed description of the changes made (as a word file).
- a word file of the manuscript text.
- individual production quality figure files (one file per figure)
- a complete author checklist, which you can download from our author guidelines

(<https://www.embopress.org/page/journal/14602075/authorguide>).

- Expanded View files (replacing Supplementary Information)

We realize that it is difficult to revise to a specific deadline. In the interest of protecting the conceptual advance provided by the work, we recommend a revision within 3 months (15th Jan 2024). Please discuss the revision progress ahead of this time with the editor if you require more time to complete the revisions.

Referee #1:

Summary

In this manuscript Stefanakis et al. developed tools to enable them to analyze the transcriptome of GLR glial cells. This analysis identified LET-381/FoxF and UNC-30/Pitx2 as key regulators of GLR fate specification. Further, they found that LET-381/FoxF and UNC-30/Pitx2, when expressed together, are sufficient to drive GLR fate in other cells. Finally, genetic ablation studies showed that the presence of GLR cells is required for locomotion and sensitivity to NaCl exposure. Together, the authors used standard *C. elegans* techniques to provide new knowledge of GLR cell specification and function that provides a platform for further dissection of glial cell specification and function.

Major concerns

- 1) I have a major concern regarding the functional data. The GLR genetic ablation strain shows robust effects on worm locomotion and salt sensitivity. However, these defects are likely caused by the disruption of neuronal positioning and signalling in the head due to lack of GLR glia. The authors examine the *let-381(ns1026)* mutant and find a very limited change in behavior compared to the ablation strain and suggest that this is due to defective GLR signaling. For the authors to suggest a non-structural effect they should use the glia specific *let-381 AID* strain to robustly remove LET-381 protein late in development and measure the behavioral phenotypes. Also, is there a reason why the authors did not investigate NaCl sensitivity with *let-381 loss*?
- 2) The authors state that *unc-30* acts autonomously in the GLR cells but no cell-specific rescue or AID experiments were performed to confirm this.
- 3) The authors state that certain genes e.g. *let-381::GFP* are expressed in GLR glia precursors. However, it is unclear how these cells were identified in the embryo. Did the authors use a reporter gene to categorically identify them or by some other means?

Minor concerns

Expression of *let-381::GFP* in coelomocytes (mentioned on page 7) is not shown.

The authors mention that LET-381 protein is rapidly degraded by K-NAA but no temporal analysis of LET-381 depletion is shown. How do the authors know that the depletion was rapid?

Page 12 - 'How might LET-381 and UNC-30 interact?' I suggest changing this sentence as it suggests that the authors were investigating a physical interaction between these two proteins.

Page 13 - 'Remarkably, the 169 bp deletion allele, *unc-30(ns998)*,' It would be helpful here if the authors state that this deletion remove part of intron 5 that houses the *let-381* motifs.

Referee #2:

The developmental program that leads to the formation and differentiation of mesodermally derived cells of the nervous system, such as microglia, is poorly understood. The authors of this manuscript investigate the transcription factors required for the differentiation of *C. elegans* GLR cells, six glial like cells found in the nematode's nervous system. These cells derive from the MS blastomere, which primarily generates body-wall and pharyngeal muscle, and thus can be considered analogous to mesodermally derived mammalian glia. Using RNA sequencing, knockouts and knockdowns, as well as GFP reporters, and degradation constructs, the authors identify two key transcription factors in the development of GLR cells: *let-381* and *unc-30*. The findings reported here are novel and interesting. The experimental design is rigorous, the experiments are well controlled, and the conclusions are supported by the data. The manuscript would benefit from a few additional experiments to tighten up the conclusions.

Major points:

1. The authors should identify a stage-specific promoter that functions during bean stage 200-400 mins to knock down *let-381* during this specific developmental stage. If their hypothesis about *let-381* controlling GLR versus muscle fate is correct, they should be able to reproduce their results with knockdown during this specific time window.
2. The authors find by deleting the *let-381* motif of *let-381* regulated genes, including *let-381* itself, leads to loss of expression of the gene. It would further strengthen their conclusions, if increase of expression could be observed following addition of multiple *let-381* motifs in any of the studied genes.
3. The authors should tone down their conclusions about lack of effect on morphology if the nerve ring is not displaced, since they have not looked at any more microscopic changes, including changes in the morphology of the individual neurons.

Minor points:

1. The authors should comment on why they focused on *let-381* and *unc-30*, instead of other transcription factors expressed in GRL cells.
2. Can the authors comment on potential effects of *let-381* on the germline?
3. Fig.1A, please check the labeling of the GLR cells (in yellow background). the label GLRVL/GLRL is repeated.
4. Figure EV5B, please add individual data points to the bar graphs.

Referee #3:

Transcriptional control of glial cell fate is poorly understood. This is an impressive and well executed study that focuses on this important knowledge gap by leveraging the strengths of the *C. elegans* model. The study focuses on a population of six mesodermally-derived glial cells, called GLR, whose development and function is poorly understood. Through RNA-Seq, the study describes the adult GPR transcriptome, raising the hypothesis that *C. elegans* GLR cells have merged astrocytic and endothelial functions. Next, the authors identify two highly conserved transcription factors, *LET-381/FoxF* and *UNC-30/PITX*, as key players in GLR cell fate development and maintenance. Through temporally-controlled manipulations, GLR-specific RNAi, and elegant CRISPR/Cas9 mutagenesis, the study uncovers a hierarchical gene regulatory network that provides deep mechanistic insights into the transcriptional mechanisms controlling glial cell fate (e.g., *LET-381/FoxF* acts directly on its effector genes, continuous requirement for *LET-381/FoxF*, *unc-30/PITX* has a dual role downstream of *LET-381/FoxF*). Conceptually, the study breaks new ground by identifying the first terminal selector-type transcription factors for glia. Overall, the paper is well written, the experiments are executed at a high level of rigor, and all conclusions are supported by the data at hand. To my eye, no additional experimentation is needed, as the paper provides very strong data to support its main conclusions in 10 main figures, 5 EV figures, and 3 Appendix S1-3 figures.

I only have minor suggestions, as indicated below:

1. The current title does not include any information on the gene regulatory network (e.g., multiple effector genes are identified) nor the hierarchical relationship uncovered (*unc-30* is a target of *LET-381*). This missed opportunity could be mitigated by simply updating the title.
2. The claim that GLR cells molecularly "resemble" mammalian astrocytes and endothelial cells is a very attractive hypothesis, mentioned in abstract and throughout. Instead of simply calling out a few genes (e.g., *snf-11*, *gbb-1*, *delp-1*, *gei-1*) in page 6, can the authors provide a more direct comparison of GO terms in *C. elegans* GLR, mammalian astrocytes and mammalian endothelial cells? Assigning cell type orthology is undoubtedly very tricky but percentages of gene families expressed in these cell types (or any other direct comparison of enriched genes) may be informative.
3. Along the same lines, the principle of "compression" is very interesting, i.e., GLR cells merge astrocytic and endothelial functions. This idea is also brought up in another *C. elegans* cell type (excitatory motor neurons). Hence, that study (PMID: 29360035) could be cited.
4. In page 8, it is stated that GLR glia are not generated in *let-381* mutants and some presumptive GLR glia acquire a muscle

fate. The authors describe this as specification defect, but it seems that there are two defects: no GLR cells are generated (hence a gliogenesis defect) plus, some presumptive GLR cells are indeed generated but adopt a muscle fate (hence a specification defect). If that is the case, some clarification on this would help the reader. Also, figure 10 could include the dual role of LET-381 (promote GLR fate at the expense of muscle fate), although it is not clear whether repression of muscle fate by LET-381 occurs in GLR dividing progenitors or postmitotic GLR cells.

5. A clear hypothesis on how GLR affect locomotion in non-structural ways could be discussed. GABA or neuropeptides involved in this?
6. Because a side-by-side comprehensive molecular profiling of GLR cells in *let-381* and *unc-30* mutants has not been performed, the claim that UNC-30 has a limited role on GLR gene expression should be toned down.
7. The Results could describe in more detail the differential effects on Dorsal GLRs versus V and L cells in *unc-30* mutants, as these effects suggest the involvement of additional, yet-to-be-identified transcription factors that participate in the GLR gene regulatory network.

RESPONSES TO EDITORIAL AND REVIEWER COMMENTS

We would like to thank the reviewers and the editor for the insightful comments on the manuscript. We have now addressed all the comments raised with either new experiments, and/or by changing the text. We believe that the manuscript is now considerably strengthened. Below are our point-by-point responses:

Reviewer #1

Summary

In this manuscript Stefanakis et al. developed tools to enable them to analyze the transcriptome of GLR glial cells. This analysis identified LET-381/FoxF and UNC-30/Pitx2 as key regulators of GLR fate specification. Further, they found that LET-381/FoxF and UNC-30/Pitx2, when expressed together, are sufficient to drive GLR fate in other cells. Finally, genetic ablation studies showed that the presence of GLR cells is required for locomotion and sensitivity to NaCl exposure. Together, the authors used standard *C. elegans* techniques to provide new knowledge of GLR cell specification and function that provides a platform for further dissection of glial cell specification and function.

We thank this reviewer for recognizing the value of our study.

Major concerns

1) I have a major concern regarding the functional data. The GLR genetic ablation strain shows robust effects on worm locomotion and salt sensitivity. However, these defects are likely caused by the disruption of neuronal positioning and signalling in the head due to lack of GLR glia. The authors examine the *let-381(ns1026)* mutant and find a very limited change in behavior compared to the ablation strain and suggest that this is due to defective GLR signaling. For the authors to suggest a non-structural effect they should use the glia specific *let-381 AID* strain to robustly remove LET-381 protein late in development and measure the behavioral phenotypes. Also, is there a reason why the authors did not investigate NaCl sensitivity with *let-381* loss?

We thank the reviewer for this excellent suggestion. We tested the *let-381::AID* strain as proposed by the reviewer: L4 animals were exposed to KNAA for 24 hours and motor behavior was then assessed. As we show in the revised Figure EV5, similar motor behavior defects are observed as in the *let-381* autoregulatory mutant. This finding further supports and strengthens our previous conclusions.

Also, is there a reason why the authors did not investigate NaCl sensitivity with *let-381* loss?

As requested by the reviewer, we now tested this. As shown in the revised Figure 9L, neither *let-381* autoregulatory mutant nor the *let-381::AID* knockdown result in hypersensitivity to high salt. We can only speculate that as opposed to GLR ablation, the GLR glia processes are still physically present in these mutants, and may therefore constitute enough of a physical barrier around the nerve ring, to protect it from the effects of sudden salt concentration shifts. We discuss this possibility in the revised text.

2) The authors state that *unc-30* acts autonomously in the GLR cells but no cell-specific rescue or AID experiments were performed to confirm this.

Our statement that *unc-30* acts cell autonomously in GLR glia is based on our finding that the 169 bp intron deletion of *unc-30*, which abolishes endogenous *unc-30::gfp* expression only in GLR glia, phenocopies the effect of the null allele on GLR gene expression and morphology. Therefore, non-autonomous function is highly unlikely. We have revised the text to make this point more clear.

Nonetheless, given the reviewer comment, we attempted to test cell autonomously using an independent method. We could not perform a simple rescue study, because expression of all our GLR-specific drivers is greatly reduced or absent in *unc-30* mutants, prohibiting their use for GLR-specific rescues. We therefore performed mosaic analysis using unstable extrachromosomal arrays. These arrays carry wild-type copies of the *unc-30* locus (*unc-30* fosmid) that rescue *unc-30(e191)* defects (Fig. EV3A) as well as lineage marker transgenes (*rab-3prom1::2xnl::tagrfp*, panneuronal, primarily AB lineage; *unc-122prom::rfp*, coelomocytes, MS lineage). Mosaic animals carrying the extrachromosomal array in the MS lineage and NOT in the AB lineage express RFP only in coelomocytes and six MS-derived pharyngeal neurons I3, I4, I6, M1, M4 and M5, easily distinguished by their stereotypic position. Animals carrying the extrachromosomal array in the AB lineage and NOT the MS lineage have broad neuronal expression (apart from the six MS-derived neurons) and no coelomocyte expression.

We found that *unc-30(e191)* null mutants carrying the extrachromosomal array in the MS lineage and not in the AB lineage have normal GLR glia development. By contrast, presence of the array in the AB lineage only does not rescue GLR defects. Because *unc-30* is only expressed in a few AB-lineage derived neurons and in the MS-lineage derived GLR glia, our findings provide additional evidence that *unc-30* acts cell autonomously to control GLR gene expression and morphology. These data are now found in Appendix Fig. 5, and discussed in the text.

3) The authors state that certain genes e.g. *let-381::GFP* are expressed in GLR glia precursors. However, it is unclear how these cells were identified in the embryo. Did the authors use a reporter gene to categorically identify them or by some other means?

We thank the reviewer for pointing this out. In a previously published study, lineaging of a *let-381::gfp* reporter during embryonic development shows *let-381* expression in GLR glia precursors. We now clarify this point in the manuscript.

Minor concerns

Expression of *let-381::GFP* in coelomocytes (mentioned on page 7) is not shown.

We added an image showing expression in coelomocytes in Appendix Figure S2A.

The authors mention that LET-381 protein is rapidly degraded by K-NAA but no temporal analysis of LET-381 depletion is shown. How do the authors know that the depletion was rapid?

This is a good point. To address it, we performed a temporal analysis of LET-381 depletion, showing that AID tagged LET-381 is degraded within 2 hours of K-NAA exposure in both L1 and L4 animals (Appendix Figure S4).

Page 12 - 'How might LET-381 and UNC-30 interact?' I suggest changing this sentence as it suggests that the authors were investigating a physical interaction between these two proteins.

We have changed this sentence to now refer to genetic interactions between the two transcription factor genes.

Page 13 - 'Remarkably, the 169 bp deletion allele, *unc-30(ns998)*,' It would be helpful here if the authors state that this deletion remove part of intron 5 that houses the *let-381* motifs.

We have added the suggested statement.

Reviewer #2

The developmental program that leads to the formation and differentiation of mesodermically derived cells of the nervous system, such as microglia, is poorly understood. The authors of this manuscript investigate the transcription factors required for the differentiation of *C. elegans* GLR cells, six glial like cells found in the nematode's nervous system. These cells derive from the MS blastomere, which primarily generates body-wall and pharyngeal muscle, and thus can be considered analogous to mesodermically derived mammalian glia. Using RNA sequencing, knockouts and knockdowns, as well as GFP reporters, and degradosome constructs, the authors identify two key transcription factors in the development of GLR cells: *let-381* and *unc-30*. The findings reported here are novel and interesting. The experimental design is rigorous, the experiments are well controlled, and the conclusions are supported by the data. The manuscript would benefit from a few additional experiments to tighten up the conclusions.

We thank this reviewer for recognizing the value of our study.

Major points:

1. The authors should identify a stage-specific promoter that functions during bean stage 200-400 mins to knock down *let-381* during this specific developmental stage. If their hypothesis about *let-381* controlling GLR versus muscle fate is correct, they should be able to reproduce their results with knockdown during this specific time window.

This experiment is a good suggestion that could provide additional support to our current model. Unfortunately, perhaps because cell-specific promoters are exceedingly rare early in embryogenesis (most gene expression appears to be combinatorially regulated at this stage), we have not identified a promoter for stage and lineage specific *let-381* knockdown in the early embryo. Nonetheless, we believe that our finding that extra muscle cells are present in *let-381* mutants, and the observation that GLR sister cells are muscle cells, plausibly suggests a cell fate transformation. We have now worded the text to convey the argument more clearly.

2. The authors find by deleting the *let-381* motif of *let-381* regulated genes, including *let-381* itself, leads to loss of expression of the gene. It would further strengthen their conclusions, if increase of expression could be observed following addition of multiple *let-381* motifs in any of the studied genes.

We agree with the reviewer. Indeed, in the original manuscript we reported additive effects of multiple motifs on gene expression levels in an *in vivo* context. Specifically, we examined the *hlh-1* gene which contains two LET-381 binding sites. As we reported, mutation of one of the motifs reduces *hlh-1::gfp* expression in GLR glia, while mutation of both motifs results in complete loss of *hlh-1::gfp* expression (Figure 5D), demonstrating that having multiple *let-381* motifs increases gene expression. Similar results were obtained by mutagenizing the first and second *let-381* motifs found in the fifth intron of the *unc-30* locus (Fig 7G).

3. The authors should tone down their conclusions about lack of effect on morphology if the nerve ring is not displaced, since they have not looked at any more microscopic changes, including changes in the morphology of the individual neurons.

Thank you for this comment. We agree that the possibility of structural defects of lesser magnitude cannot be excluded and have revised the text accordingly.

Minor points:

1. The authors should comment on why they focused on *let-381* and *unc-30*, instead of other transcription factors expressed in GRL cells.

We used available mutants and/or RNAi to test the roles of the top seven TFs in Table EV1 and only observed defects in *let-381* and *unc-30*. This is now clarified in the Table EV1 legend.

2. Can the authors comment on potential effects of *let-381* on the germline?

As we mention in the manuscript, a few *let-381(gk302)* null mutant animals escape lethality to become sterile adults. It is known from previous studies that *let-381* mutant animals lack the post-embryonically born coelomocytes, with extra sex muscle cells being generated in their place. Germline development depends on interactions with somatic gonad cells, it is thus possible that sterility of *let-381* mutant worms is an indirect effect due to somatic gonad/sex muscle specification defects. This is now noted in the manuscript.

3. Fig.1A, please check the labeling of the GLR cells (in yellow background). the label GLRVL/GLRL is repeated.

Thank you for noticing this. We have now fixed the label.

4. Figure EV5B, please add individual data points to the bar graphs.

We have added data points as requested.

Reviewer #3

Transcriptional control of glial cell fate is poorly understood. This is an impressive and well executed study that focuses on this important knowledge gap by leveraging the strengths of the *C. elegans* model. The study focuses on a population of six mesodermally-derived glial cells, called GLR, whose development and function is poorly understood. Through RNA-Seq, the study describes the adult GPR transcriptome, raising the hypothesis that *C. elegans* GLR cells have merged astrocytic and endothelial functions. Next, the authors identify two highly conserved transcription factors, LET-381/FoxF and UNC-30/PITX, as key players in GLR cell fate development and maintenance. Through temporally-controlled manipulations, GLR-specific RNAi, and elegant CRISPR/Cas9 mutagenesis, the study uncovers a hierarchical gene regulatory network that provides deep mechanistic insights into the transcriptional mechanisms controlling glial cell fate (e.g., LET-381/FoxF acts directly on its effector genes, continuous requirement for LET-381/FoxF, unc-30/PITX has a dual role downstream of LET-381/FoxF). Conceptually, the study breaks new ground by identifying the first terminal selector-type transcription factors for glia. Overall, the paper is well written, the experiments are executed at a high level of rigor, and all conclusions are supported by the data at hand. To my eye, no additional experimentation is needed, as the paper provides very strong data to support its main conclusions in 10 main figures, 5 EV figures, and 3 Appendix S1-3 figures.

We thank this reviewer for recognizing the value of our study.

I only have minor suggestions, as indicated below:

1. The current title does not include any information on the gene regulatory network (e.g., multiple effector genes are identified) nor the hierarchical relationship uncovered (unc-30 is a target of LET-381). This missed opportunity could be mitigated by simply updating the title.

We thank the reviewer for the suggestion. We have now included more information on the identified regulatory network in the manuscripts title.

2. The claim that GLR cells molecularly "resemble" mammalian astrocytes and endothelial cells is a very attractive hypothesis, mentioned in abstract and throughout. Instead of simply calling out a few genes (e.g., snf-11, gbb-1, delp-1, gei-1) in page 6, can the authors provide a more direct comparison of GO terms in *C. elegans* GLR, mammalian astrocytes and mammalian endothelia cells? Assigning cell type orthology is undoubtedly very tricky but percentages of gene families expressed in these cell types (or any other direct comparison of enriched genes) may be informative.

We very much agree with the reviewer that a broader molecular comparison would be useful here. However, as the reviewer also notes, cross-species transcriptome comparisons are very difficult to perform, as identifying the correct cognate genes is often impossible, and gene expression levels or even ranking of gene expression is unlikely to be preserved. We attempted the GO analysis proposed by the reviewer, but it was not able to provide any additional insight. We thus opted to leave the comparison as is.

3. Along the same lines, the principle of "compression" is very interesting, i.e., GLR cells merge astrocytic and endothelial functions. This idea is also brought up in another *C. elegans* cell type (excitatory motor neurons). Hence, that study (PMID: 29360035) could be cited.

We thank the reviewer for bringing this study to our attention. This is another great example of “compression” in *C. elegans*, now cited in our manuscript.

4. In page 8, it is stated that GLR glia are not generated in *let-381* mutants and some presumptive GLR glia acquire a muscle fate. The authors describe this as specification defect, but it seems that there are two defects: no GLR cells are generated (hence a gliogenesis defect) plus, some presumptive GLR cells are indeed generated but adopt a muscle fate (hence a specification defect). If that is the case, some clarification on this would help the reader. Also, figure 10 could include the dual role of LET-381 (promote GLR fate at the expense of muscle fate), although it is not clear whether repression of muscle fate by LET-381 occurs in GLR dividing progenitors or postmitotic GLR cells.

We do not know the exact mechanism of extra body wall muscle production in the *let-381* null mutants. GLR glia are indeed not specified as suggested by the absence of any tested GLR marker and by the anterior positioning of the nerve ring. It is possible that in the absence of *let-381*, GLR lineages adopt sister MS-lineage fates producing body wall muscle instead. We have now rephrased this in the manuscript, hoping that it is clearer.

Regarding Figure 10, we believe it already shows the dual role of LET-381 in promoting GLR fate specification and repressing alternative muscle fate.

5. A clear hypothesis on how GLR affect locomotion in non-structural ways could be discussed. GABA or neuropeptides involved in this?

Based on their anatomy and gene expression, GLR glia could affect motor output in several ways. Some hypotheses are added in the Discussion.

6. Because a side-by-side comprehensive molecular profiling of GLR cells in *let-381* and *unc-30* mutants has not been performed, the claim that UNC-30 has a limited role on GLR gene expression should be toned down.

Although a transcriptome comparison between wild type, *let-381* and *unc-30* mutants may clarify the magnitude of effect of these genes on GLR gene expression, we believe, nonetheless, that our statement that *unc-30* has a more restricted effect on gene expression than *let-381* is very likely correct. For one, *unc-30* loss affects only lateral and ventral but not dorsal GLR glia, unlike *let-381* loss. In addition, for several genes we examined in *unc-30* mutants, expression is reduced and not abolished as it is in *let-381* animals. We have now rephrased the relevant sentence in the manuscript to clarify the point.

7. The Results could describe in more detail the differential effects on Dorsal GLRs versus V and L cells in *unc-30* mutants, as these effects suggest the involvement of additional, yet-to-be-identified transcription factors that participate in the GLR gene regulatory network.

Thank you for this suggestion. We now emphasize the difference between dorsal and ventra/lateral GLR glia in the Results section.

Dear Dr. Shaham,

Thank you for submitting a revised version of your manuscript. Your study has now been seen by all original referees, who find that their previous concerns have been addressed and now recommend acceptance of the manuscript.

There now remain a few editorial points that need addressing before I can extend acceptance of the manuscript.

With best wishes,

Ieva

We realize that it is difficult to revise to a specific deadline. In the interest of protecting the conceptual advance provided by the work, we recommend a revision within 3 months (17th Apr 2024). Please discuss the revision progress ahead of this time with the editor if you require more time to complete the revisions.

Referee #1:

My original concerns have now been resolved.

Referee #2:

I am satisfied with how the authors addressed my concerns. I have no further comments. I now deem the manuscript suitable for publication.

Referee #3:

The authors have addressed all issues raised by the three reviewers. The new experiments and text changes have strengthened even more this work, which provides a significant contribution to the field of developmental biology.

Senior Scientific Editor
The EMBO Journal
Meyerhofstrasse 1
D-69117 Heidelberg
Tel: +4962218891309
i.gailite@embojournal.org

We realize that it is difficult to revise to a specific deadline. In the interest of protecting the conceptual advance provided by the work, we recommend a revision within 3 months (17th Apr 2024). Please discuss the revision progress ahead of this time with the editor if you require more time to complete the revisions.

Referee #1:

My original concerns have now been resolved.

Referee #2:

I am satisfied with how the authors addressed my concerns. I have no further comments. I now deem the manuscript suitable for publication.

Referee #3:

The authors have addressed all issues raised by the three reviewers. The new experiments and text changes have strengthened even more this work, which provides a significant contribution to the field of developmental biology.

Dear Dr. Shaham,

Thank you for addressing the final editorial issues. I am now pleased to inform you that your manuscript has been accepted for publication.

Before we forward your manuscript to our publishers, I would like to propose a couple of minor changes in the article title, abstract and synopsis, mainly aimed at increasing the accessibility of the study to our more general audience. I have also written a short blurb that will accompany the title of your manuscript in our online table of contents. Please take a look at the text below and in the attached manuscript text file and let me know if any corrections are necessary.

Title:
Development of mesodermally-derived *C. elegans* glia depends on terminal selector genes LET-381/FoxF and UNC-30/Pitx2

Blurb:
GLR glia that envelop the *C. elegans* central nervous system show a mixed astrocyte/endothelial cell identity and regulate motor behavior.

Synopsis:
Mesodermal glia have crucial functions in the nervous system, although the mechanisms regulating their development remain elusive. Here, transcriptomic and mutational analyses identify a gene regulatory network required for specification and maintenance of mesodermally-derived GLR glia in *C. elegans*.

- In *let-381/FoxF* null mutants, GLR glia are not specified and some GLR lineages adopt sister muscle lineage fates instead.
- LET-381/FoxF acts as an autoregulatory terminal selector to maintain GLR gene expression via a common cis-regulatory motif.
- UNC-30/Pitx2 represses head mesodermal cell fate in GLR glia and regulates GLR morphology and gene expression.
- GLR glia ablation results in severe motor behavior defects and salt hypersensitivity.

If you have any questions, please do not hesitate to contact the Editorial Office. Thank you again for this contribution to The EMBO Journal and congratulations on a great study!

Best wishes,

leva

leva Gailite, PhD
Senior Scientific Editor
The EMBO Journal
Meyerhofstrasse 1
D-69117 Heidelberg
Tel: +4962218891309
i.gailite@embojournal.org
